# Signal Processing Meets SGD: From Momentum to Filter

## Abstract

In deep learning, stochastic gradient descent (SGD) and its momentum-based variants are widely used for optimization, but they typically suffer from slow convergence. Conversely, existing adaptive learning rate optimizers speed up convergence but often compromise generalization. To resolve this issue, we propose a novel optimization method designed to accelerate SGD's convergence without sacrificing generalization. Our approach reduces the variance of the historical gradient, improves first-order moment estimation of SGD by applying Wiener filter theory, and introduces a time-varying adaptive gain. Empirical results demonstrate that SGDF (SGD with Filter) effectively balances convergence and generalization compared to state-of-the-art optimizers. The code is available at https://anonymous.4open.science/r/SGDF-Optimizer/.

## 1 Introduction

During the training process, the optimizer serves as a critical component of the model. It refines and adjusts model parameters to ensure that the model can recognize underlying data patterns. Beyond updating weights, the optimizer's role includes strategically navigating complex loss landscapes (Du & Lee, 2018) to locate regions that offer the best generalization (Keskar et al., 2022). The chosen optimizer significantly impacts training efficiency, influencing model convergence speed, generalization performance, and resilience to data distribution shifts (Bengio & Lecun, 2007). A poor optimizer choice can result in suboptimal convergence or failure to converge, whereas a suitable one can accelerate learning and ensure robust performance (Ruder, 2016). Thus, continually refining optimization algorithms is essential for enhancing the capabilities of machine learning models.

Meanwhile, Stochastic Gradient Descent (SGD) (Monro, 1951) and its variants, such as momentum-based SGD (Sutskever et al., 2013), Adam (Kingma & Ba, 2014), and RMSprop (Hinton et al., 2012), have secured prominent roles. Despite their substantial contributions to deep learning, these methods have inherent drawbacks. They primarily exploit first-order moment estimation and frequently overlook the pivotal influence of historical gradients on current parameter adjustments. Consequently, they can result in training instability or poor generalization (Chandramoorthy et al., 2022), especially with high-dimensional, non-convex loss functions common in deep learning (Goodfellow et al., 2016). Such characteristics render adaptive learning rate methods prone to entrapment in sharp local minima, which can significantly impair the model's generalization capability (Zhang et al., 2021). Various Adam variants (Chen et al., 2018a; Liu et al., 2019; Luo et al., 2019; Zhuang et al., 2020) aim to improve optimization and enhance generalization performance by adjusting the adaptive learning rate. Although these variants have achieved some success, they still have not completely resolved the issue of generalization loss.

To achieve an effective trade-off between convergence speed and generalization capability (Geman et al., 2014), this paper introduces a novel optimization method called SGDF (SGD with Filter). SGDF incorporates filter theory from signal processing to enhance first-moment estimation, balancing historical and current gradient estimates. Through its adaptive weighting mechanism, SGDF precisely adjusts gradient estimates throughout the training process, thereby accelerating model convergence while preserving generalization ability.

Initial evaluations demonstrate that SGDF surpasses many traditional adaptive learning rate and variance reduction optimization methods across various benchmark datasets, particularly in terms of accelerating convergence and maintaining generalization. This indicates that SGDF successfully

navigates the trade-off between speeding up convergence and preserving generalization capability. By achieving this balance, SGDF offers a more efficient and robust optimization option for training deep learning models.

The main contributions of this paper can be summarized as follows:

- We introduce SGDF, an optimizer that integrates historical and current gradient data to compute the gradient's variance estimate, addressing the slow convergence of the vanilla SGD method.
- We theoretically analyze the benefits of SGDF in terms of generalization (Sec. 3.3 )and convergence (Sec. 3.4), and empirically verify the effectiveness of SGDF (Sec. 4).
- We employ first-moment filter estimation in SGDF, which can also significantly enhance the generalization capacity of adaptive optimization algorithms (*e.g.*, Adam) (Sec. 4.4), surpassing traditional momentum strategies.

## 2 PRELIMINARY ANALYSIS

### 2.1 PRELIMINARIES

**Batch Normalization:** Batch Normalization (BN) (Ioffe & Szegedy, 2015) is widely used to normalize and rescale mini-batch data, reducing internal covariate shift and stabilizing gradient distributions. BN helps mitigate gradient vanishing/exploding, improving convergence speed and stability. The core BN operation is $\hat{x}^{(k)} = \dfrac{x^{(k)} - \mu_B}{\sqrt{\sigma_B^2 + \epsilon}}$, where $\mu_B$ and $\sigma_B^2$ are the mini-batch mean and variance, and $\epsilon$ is for numerical stability. The normalized values are rescaled as $y^{(k)} = \gamma \hat{x}^{(k)} + \beta$.

**Signal Processing:** Filters in signal processing are used to manipulate the frequency components of a signal, typically to reduce noise or enhance specific features. One common example is the Low Pass Filter, which smooths high frequency fluctuations by applying an exponential moving average. (Liu et al., 2019) generalized that the first-moment (momentum) of adaptive-based optimizers can be expressed as $\phi(x_1, \cdots, x_t) = \dfrac{(1 - \beta_1) \sum_{i=1}^{t} \beta_1^{t-i} x_i}{1 - \beta_1^t}$, where $\beta_1$ is the smoothing factor controlling the influence of past values in the exponential moving average. To differentiate this from the standard momentum method discussed in later sections (Sutskever et al., 2013), we refer to this exponential moving average form of SGD as SGD-LPF (Low Pass Filter) in this section. Another important filter is the Wiener Filter (Wiener, 1950), which minimizes the mean square error between an estimated signal and the true signal by filtering out noise. Unlike a simple low-pass filter, the Wiener Filter has time-varying gain, adapting its response dynamically based on the characteristics of the signal and noise. The Wiener filter's frequency response is given by $H(f) = \dfrac{S_{xx}(f)}{S_{xx}(f) + S_{nn}(f)}$, where $S_{xx}(f)$ is the power spectral density of the signal and $S_{nn}(f)$ is the power spectral density of the noise. This adaptive nature allows for more accurate signal recovery by optimally balancing noise reduction and signal preservation.

### 2.2 GRADIENT ANALYSIS

We performed a series of experiments to evaluate the overall performance of VGG networks (Simonyan & Zisserman, 2014) trained using different techniques with SGD. We first compared Vanilla SGD, SGD-BN (trained using a VGG with BN), SGD-LPF, and the Wiener Filter applied in our proposed SGDF algorithm in terms of overall performance. Afterward, we observed the impact of these techniques on the gradient distributions within the feature layers.

From the Fig. 1, it is clear that the VGG trained without BN using vanilla SGD exhibits lower accuracy and slower convergence in both the training and testing phases. In contrast, the VGG with BN significantly improves both convergence speed and accuracy. SGD-LPF helps smooth the gradient fluctuations and accelerates convergence, but still results in lower performance compared to the BN-enhanced network. However, the Wiener Filter SGDF algorithm achieves the best performance, with both training and testing accuracies significantly surpassing other methods, while also converging faster and more stably throughout the training process.

We recorded the gradient values of the feature layers during the first 100 iterations for each algorithm. Using kernel density estimation, we sampled these gradients to generate PDF curves, which are presented in Fig. 2. In the VGG network without BN, the gradient distributions of the feature layers show significant instability. **SGD:** As Fig. 2 (a) shown, the gradient of different layers fluctuates greatly and is unevenly distributed, which causes the network to oscillate during the training process and makes it difficult to converge stably. **SGD-BN:** In the VGG network with BN, on the other hand, the gradient variance is significantly reduced as seen in Fig. 2 (c), and the gradient distribution becomes smoother and more concentrated. **SGD-LPF:** Similarly, the Fig. 2(d) shows that SGD-LPF effectively smooths the gradient fluctuations through the exponential moving average. However, due to the fixed weighting coefficient, there is still a certain degree of gradient shift during some iterations, which can lead to systematic bias in the gradient update direction during training, ultimately preventing the performance from surpassing that of the BN-enhanced network. **SGD-WF:** Finally, Fig. 2 (b) presents the gradient distribution of the VGG network trained with the Wiener-filtered SGDF algorithm. Compared to other methods, SGDF produces a gradient distribution as concentrated as BN, with less noise and no gradient shift. This improvement leads to a more stable training process and better convergence across all layers.

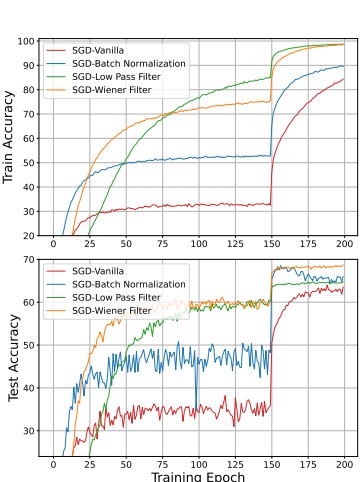

Figure 1: Training of VGG on the CIFAR-100 dataset.

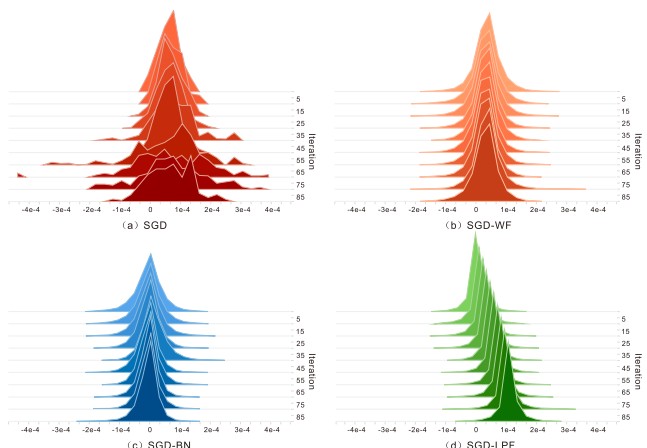

Figure 2: The gradient histogram of the VGG on the CIFAR-100 dataset. The x-axis is the gradient value and the height is the frequency. SGD trains the VGG without BN, the variance of the gradient fluctuates dramatically and the update is unstable.

## 3 METHOD

We can find from the previous section that reducing the variance can improve the performance of SGD. However, previous variance reduction techniques (Defazio et al., 2014; Johnson & Zhang, 2013; Schmidt et al., 2017) have in turn impaired the generalization ability of SGD, and we introduce SGDF in this section and highlight in 3.3 why our method does not impair generalization.

### 3.1 SGDF GENERAL INTRODUCTION

In algorithm 1, $s_t$ serves as a key indicator, calculated as the exponential moving average of the squared difference between the current gradient $g_t$ and its momentum $m_t$, acting as a marker for gradient variation with weight-adjusted by $\beta_2$. (Zhuang et al., 2020) first proposed the calculation of $s_t$, which is utilized for estimating the fluctuation variance of the stochastic gradient. We derived a correction factor $(1 - \beta_1)(1 - \beta_1^{2t})/(1 + \beta_1)$ under the assumption that $m_t$ and $g_t$ are independently and identically distributed (i.i.d.), to accurately estimate the variance of $m_t$ using $s_t$. Fig. 12 compares performances with and without the correction factor, showing superior results with correction. For the derivation of the correction factor, please refer to Appendix A.2.

---

**Algorithm 1:** SGDF, Wiener Filter Estimate Gradient. All operations are element-wise.

---

**Input:** $\{\alpha_t\}_{t=1}^T$: step size, $\{\beta_1, \beta_2\}$: attenuation coefficient, $\theta_0$: initial parameter, $f(\theta)$:
  stochastic objective function
**Output:** $\theta_T$: resulting parameters.
Init: $m_0 \leftarrow 0$, $s_0 \leftarrow 0$
**while** $t = 1$ to $T$ **do**
  $g_t \leftarrow \nabla f_t(\theta_{t-1})$ (Calculate Gradients w.r.t. Stochastic Objective at Timestep $t$)
  $m_t \leftarrow \beta_1 m_{t-1} + (1 - \beta_1)g_t$ (Calculate Exponential Moving Average)
  $s_t \leftarrow \beta_2 s_{t-1} + (1 - \beta_2)(g_t - m_t)^2$ (Calculate Exponential Moving Variance)
  $\widehat{m}_t \leftarrow \dfrac{m_t}{1 - \beta_1^t}, \widehat{s}_t \leftarrow \dfrac{(1 - \beta_1)(1 - \beta_1^{2t})s_t}{(1 + \beta_1)(1 - \beta_2^t)}$ (Bias Correction)
  $K_t \leftarrow \dfrac{\widehat{s}_t}{\widehat{s}_t + (g_t - \widehat{m}_t)^2}$ (Calculate Estimate Gain)
  $\widehat{g}_t \leftarrow \widehat{m}_t + K_t(g_t - \widehat{m}_t)$ (Update Gradient Estimation)
  $\theta_t \leftarrow \theta_{t-1} - \alpha_t \widehat{g}_t$ (Update Parameters)
return $\theta_T$

---

At each time step $t$, $g_t$ represents the stochastic gradient for our objective function, while $m_t$ approximates the historical trend of the gradient through an exponential moving average. The difference $g_t - m_t$ highlights the gradient's deviation from its historical pattern, reflecting the inherent noise or uncertainty in the instantaneous gradient estimate, which can be expressed as $p(g_t|\mathcal{D}) \sim \mathcal{N}(g_t; m_t, \sigma_t^2)$ (Liu et al., 2019).

SGDF utilizes the gain $K_t$, where the components of each dimension of the estimated gain range between 0 and 1, to balance the current observed gradient $g_t$ and the past corrected gradient $\widehat{m}_t$, thus optimizing the gradient estimate. This balance plays a crucial role in noisy or complex optimization scenarios, helping to mitigate noise and achieve stable gradient direction, faster convergence, and enhanced performance. The computation of $K_t$, based on $s_t$ and $g_t - m_t$, aims to minimize the expected variance of the corrected gradient $\widehat{g}_t$ for optimal linear estimation in noisy conditions. For the method derivation, please refer to Appendix A.1.

## 3.2 Fusion of Gaussian Distributions for Gradient Estimate

By fusing two Gaussian distributions, SGDF significantly reduces the variance of gradient estimates, thereby benefiting in solving complex stochastic optimization problems. In this section, we will delve into how SGDF achieves the reduction of gradient estimate variance.

The properties of SGDF ensure that the estimated gradient is a linear combination of the current noisy gradient observation $g_t$ and the first-order moment estimate $\widehat{m}_t$. These two components are assumed to have Gaussian distributions, where $g_i \sim \mathcal{N}(\mu, \sigma^2)$. Hence, their fusion by the filter naturally ensures that the fused estimate $\widehat{g}_t$ is also Gaussian.

Consider two Gaussian distributions for the momentum term $\widehat{m}_t$ and the current gradient $g_t$:

- The exponential moving average term $\widehat{m}_t$ is normally distributed with mean $\mu_m$ and variance $\sigma_m^2$, denoted as $\widehat{m}_t \sim \mathcal{N}(\mu_m, \sigma_m^2)$.
- The current gradient $g_t$ is normally distributed with mean $\mu_g$ and variance $\sigma_g^2$, denoted as $g_t \sim \mathcal{N}(\mu_g, \sigma_g^2)$.

The product of their probability density functions is given by:

$$N(\widehat{m}_t; \mu_m, \sigma_m) \cdot N(g_t; \mu_g, \sigma_g) = \frac{1}{2\pi\sigma_m\sigma_g} \exp\left(-\frac{(\widehat{m}_t - \mu_m)^2}{2\sigma_m^2} - \frac{(g_t - \mu_g)^2}{2\sigma_g^2}\right) \quad (1)$$

Through coefficient matching in the exponential terms, we obtain the new mean and variance:

$$\mu' = \frac{\sigma_g^2 \mu_m + \sigma_m^2 \mu_g}{\sigma_m^2 + \sigma_g^2} \quad \sigma'^2 = \frac{\sigma_m^2 \sigma_g^2}{\sigma_m^2 + \sigma_g^2} \quad (2)$$

The new mean $\mu'$ is a weighted average of the two means, $\mu_m$ and $\mu_g$, with weights inversely proportional to their variances. This places $\mu'$ between $\mu_m$ and $\mu_g$, closer to the mean with the smaller variance. The new standard deviation $\sigma'$ is smaller than either of the original standard deviations $\sigma_m$ and $\sigma_g$, reflecting the reduced uncertainty in the estimate due to the combination of information from both sources. This is a direct consequence of the Wiener Filter's optimality in the mean-square error sense. The proof is provided in Appendix A.3.

### 3.3 GENERALIZATION ANALYSIS OF THE VARIANCE LOWER BOUND

In previous variance reduction techniques, variance is reduced at a rate of $\xi^{t-1}, \xi \in (0, 1)$. However, this can lower the variance to a point where it limits necessary stochastic exploration, hindering optimization. The Wiener Filter, guided by the Cramér-Rao lower bound (CRLB) (Rao, 1992), ensures a lower bound on variance. We model this advantage using the Fokker-Planck equation to highlight the optimization benefits of maintaining a variance lower bound.

**Theorem 3.1.** *Consider a system governed by the Fokker-Planck equation, describing the evolution of the probability density $P$ in parameter space. For a loss function $f(\theta)$ and a noise variance matrix $D_{ij}$ satisfying $D_i \geq C > 0$, with $C$ as the Cramér-Rao lower bound, the steady-state probability density ($\frac{\partial P}{\partial t} = 0$) is:*

$$P(\theta) = \frac{1}{Z} \exp\left(-\sum_{i=1}^{n} \frac{f(\theta)}{D_i}\right), \tag{3}$$

*where $Z$ is the normalization constant, assuming $D_{ij} = D_i \delta_{ij}$.*

The existence of a variance lower bound critically enhances the algorithm's exploration capabilities, especially in regions of the loss landscape where gradients are minimal. By preventing the probability density function from becoming unbounded, it ensures continuous exploration and increases the probability of converging to flat minima associated with better generalization properties (Yang et al., 2023). The proof of Theorem 3.1 is provided in Appendix A.4.

### 3.4 CONVERGENCE ANALYSIS IN CONVEX AND NON-CONVEX OPTIMIZATION

Finally, we provide the convergence property of SGDF as shown in Theorem 3.2 and Theorem 3.3. The assumptions are common and standard when analyzing the convergence of convex and non-convex functions via SGD-based methods (Chen et al., 2018b; Kingma & Ba, 2014; Reddi et al., 2018). Proofs for convergence in convex and non-convex cases are provided in Appendix B and Appendix C, respectively. In the convergence analysis, the assumptions are relaxed and the upper bound is reduced due to the estimation gain introduced by SGDF, promoting faster convergence.

**Theorem 3.2.** *(Convergence in convex optimization) Assume that the function $f_t$ has bounded gradients, $\|\nabla f_t(\theta)\|_2 \leq G$, $\|\nabla f_t(\theta)\|_\infty \leq G_\infty$ for all $\theta \in \mathbb{R}^d$ and distance between any $\theta_t$ generated by SGDF is bounded, $\|\theta_n - \theta_m\|_2 \leq D$, $\|\theta_m - \theta_n\|_\infty \leq D_\infty$ for any $m, n \in \{1, ..., T\}$, and $\beta_1, \beta_2 \in [0, 1)$. Let $\alpha_t = \alpha/\sqrt{t}$. SGDF achieves the following guarantee, for all $T \geq 1$:*

$$R(T) \leq \frac{D^2}{\alpha} \sum_{i=1}^{d} \sqrt{T} + \frac{2D_\infty G_\infty}{1 - \beta_1} \sum_{i=1}^{d} \|g_{1:T,i}\|_2 + \frac{2\alpha G_\infty^2 (1 + (1 - \beta_1)^2)}{\sqrt{T}(1 - \beta_1)^2} \sum_{i=1}^{d} \|g_{1:T,i}\|_2^2 \tag{4}$$

*where $R(T) = \sum_{t=1}^{T} f_t(\theta_t) - f_t(\theta^*)$ denotes the cumulative performance gap between the generated solution and the optimal solution.*

For the convex case, Theorem 3.2 implies that the regret of SGDF is upper bounded by $O(\sqrt{T})$. In the Adam-type optimizers, it's crucial for the convex analysis to decay $\beta_{1,t}$ towards zero (Kingma & Ba, 2014; Zhuang et al., 2020). We have relaxed the analysis assumption by introducing a time-varying gain $K_t$, which can adapt with variance. Moreover, $K_t$ converges with variance at the end of training to improve convergence (Sutskever et al., 2013).

**Theorem 3.3.** *(Convergence for non-convex stochastic optimization) Under the assumptions:*

- *A1 Bounded variables (same as convex). $\|\theta - \theta^*\|_2 \leq D, \ \forall \theta, \theta^*$ or for any dimension $i$ of the variable, $\|\theta_i - \theta_i^*\|_2 \leq D_i, \ \forall \theta_i, \theta_i^*$*

- *A2 The noisy gradient is unbiased. For $\forall t$, the random variable $\zeta_t$ is defined as $\zeta_t = g_t - \nabla f(\theta_t)$, $\zeta_t$ satisfy $\mathbb{E}[\zeta_t] = 0$, $\mathbb{E}\left[\|\zeta_t\|_2^2\right] \leq \sigma^2$, and when $t_1 \neq t_2$, $\zeta_{t_1}$ and $\zeta_{t_2}$ are statistically independent, i.e., $\zeta_{t_1} \perp \zeta_{t_2}$.*

- *A3 Bounded gradient and noisy gradient. At step $t$, the algorithm can access a bounded noisy gradient, and the true gradient is also bounded. i.e. $\|\nabla f(\theta_t)\| \leq G$, $\|g_t\| \leq G$, $\forall t > 1$.*

- *A4 The property of function. (1) $f$ is differentiable; (2) $\|\nabla f(x) - \nabla f(y)\| \leq L\|x - y\|$, $\forall x, y$; (3) $f$ is also lower bounded.*

*Consider a non-convex optimization problem. Suppose assumptions A1-A4 are satisfied, and let $\alpha_t = \alpha/\sqrt{t}$. For all $T \geq 1$, SGDF achieves the following guarantee:*

$$\mathbb{E}(T) \leq \frac{C_7 \alpha^2 (\log T + 1) + C_8}{2\alpha\sqrt{T}} \tag{5}$$

*where $\mathbb{E}(T) = \min_{t=1,2,\ldots,T} \mathbb{E}_{t-1}\left[\|\nabla f(\theta_t)\|_2^2\right]$ denotes the minimum of the squared-paradigm expectation of the gradient, $\alpha$ is the learning rate at the $1$-th step, $C_7$ are constants independent of $d$ and $T$, $C_8$ is a constant independent of $T$, and the expectation is taken w.r.t all randomness corresponding to $g_t$.*

Theorem 3.3 indicates that the convergence rate for SGDF in the non-convex case is $O(\log T/\sqrt{T})$, which is comparable to Adam-type optimizers (Chen et al., 2018b; Reddi et al., 2018). Note that in our derivation, the terms related to the estimated gain $K_t$ were scaled to their maximum upper bounds, simplifying the upper bound results. Importantly, we did not rely on the $\mu$-strongly convex assumption (Balles & Hennig, 2018) but used the most general smoothness assumption to obtain this convergence rate. In practice, convergence speed will improve as variance diminishes, causing $K_t$ to converge more rapidly and influencing the overall convergence rate. This reduction in the upper bound due to the convergence of variance explains why SGDF converges faster than SGD.

## 4 EXPERIMENTS

### 4.1 EMPIRICAL EVALUATION

In this study, we focus on the following tasks: **Image Classification.** We employed the VGG (Simonyan & Zisserman, 2014), ResNet (He et al., 2016), and DenseNet (Huang et al., 2017) models for image classification tasks on the CIFAR dataset (Krizhevsky et al., 2009). The major difference between these three network architectures is the residual connectivity, which we will discuss in Sec. 4.4. We evaluated and compared the performance of SGDF with other optimizers such as SGD, Adam, RAdam (Liu et al., 2019), AdamW (Loshchilov & Hutter, 2017), MSVAG (Balles & Hennig, 2018), Adabound (Luo et al., 2019), Sophia (Liu et al., 2023), and Lion (Chen et al., 2023), all of which were implemented based on the official PyTorch. Additionally, we further tested the performance of SGDF on the ImageNet dataset Deng et al. (2009) using the ResNet model. **Object Detection.** Object detection was performed on the PASCAL VOC dataset (Everingham et al., 2010) using Faster-RCNN (Ren et al., 2015) integrated with FPN. For hyper-parameter tuning related to image classification and object detection, refer to (Zhuang et al., 2020). **Image Generation.** Wasserstein-GAN (WGAN) (Arjovsky et al., 2017) on the CIFAR-10 dataset.

**Hyperparameter tuning.** Following Zhuang *et al.* (Zhuang et al., 2020), we delved deep into the optimal hyperparameter settings for our experiments. In the image classification task, we employed these settings:

- *SGDF:* We adhered to Adam's original parameter values: $\beta_1 = 0.9$, $\beta_2 = 0.999$, $\epsilon = 10^{-8}$.

- *SGD:* We set the momentum to 0.9, the default for networks like ResNet and DenseNet. The learning rate was searched in the set $\{10.0, 1.0, 0.1, 0.01, 0.001\}$.

- *Adam, RAdam, MSVAG, AdaBound:* Traversing the hyperparameter landscape, we scored $\beta_1$ values in $\{0.5, 0.6, 0.7, 0.8, 0.9\}$, probed $\alpha$ as in SGD, while tethering other parameters to their literary defaults.

- *AdamW, SophiaG, Lion:* Mirroring Adam's parameter search schema, we fixed weight decay at $5 \times 10^{-4}$; yet for AdamW, whose optimal decay often exceeds norms (Loshchilov & Hutter, 2017), we ranged weight decay over $\left\{10^{-4}, 5 \times 10^{-4}, 10^{-3}, 10^{-2}, 10^{-1}\right\}$.

- *SophiaG, Lion:* We searched for the learning rate among $\{10^{-3}, 10^{-4}, 10^{-5}\}$ and adjusted Lion's learning rate (Liu et al., 2023). Following (Liu et al., 2023; Chen et al., 2023), we set $\beta_1$=0.965, 0.9 and $\beta_2$=0.99 as the default parameters.

**CIFAR-10/100 Experiments.** We initially trained on the CIFAR-10 and CIFAR-100 datasets using the VGG, ResNet, and DenseNet models and assessed the performance of the SGDF optimizer. In these experiments, we employed basic data augmentation techniques such as random horizontal flip and random cropping (with a 4-pixel padding). To facilitate result reproduction, we provide the parameter table for this subpart in Tab. 5. The results represent the mean and standard deviation of 3 runs, visualized as curve graphs in Fig. 3.

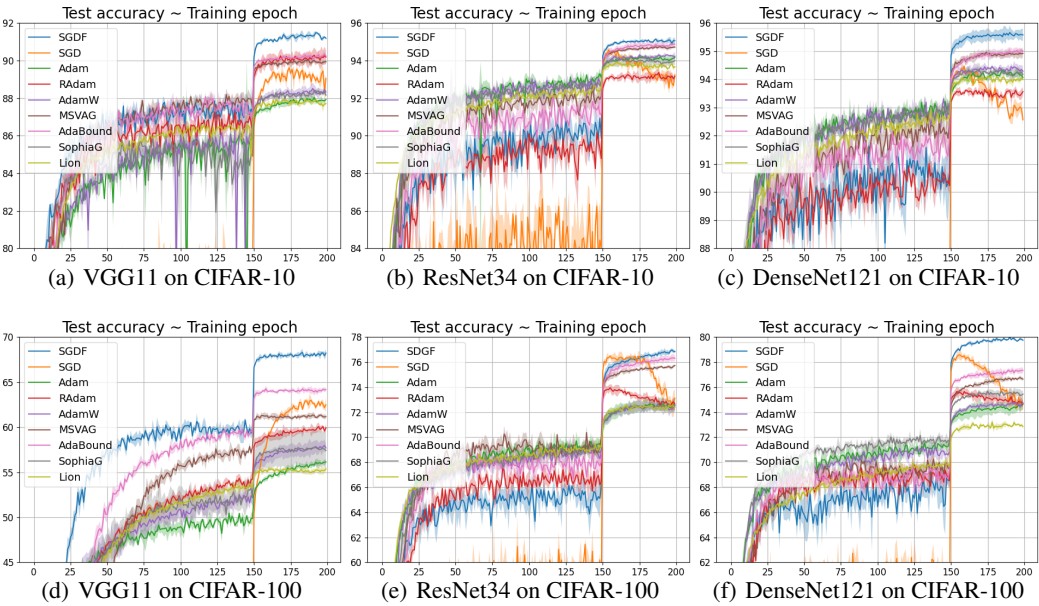

Figure 3: Test accuracy ($[\mu \pm \sigma]$) on CIFAR.

As Fig. 3 shows, that it is evident that the SGDF optimizer exhibited convergence speeds comparable to adaptive optimization algorithms. Additionally, SGDF's final test set accuracy was either better than or equal to that achieved by SGD.

**ImageNet Experiments.** We use the best-reported parameters from (Chen et al., 2018a; Liu et al., 2019). We applied basic data augmentation strategies such as random cropping and random horizontal flipping. The results are presented in Tab. 1. To facilitate result reproduction, we provide the parameter table for this subpart in Tab. 6. Detailed training and test curves are depicted in Fig. 9. Additionally, to mitigate the effect of learning rate scheduling, we employed cosine learning rate scheduling as suggested by (Chen et al., 2023; Zhang et al., 2023) and trained ResNet18, 34, and 50 models. The results are summarized in Tab. 2. Experiments on the ImageNet dataset demonstrate that SGDF has improved convergence speed and achieves similar accuracy to SGD on the test set.

Table 1: Top-1, 5 accuracy of ResNet18 on ImageNet. $*$ $\dagger$ $\ddagger$ is reported in Zhuang et al. (2020); Chen et al. (2018a); Liu et al. (2019).

| Method | SGDF | SGD | AdaBound | Yogi | MSVAG | Adam | RAdam | AdamW |
|--------|------|-----|----------|------|-------|------|-------|-------|
| Top-1 | **70.23** | 70.23[†] | 68.13[†] | 68.23[†] | 65.99[*] | 63.79[†] (66.54[‡]) | 67.62[‡] | 67.93[†] |
| Top-5 | **89.55** | 89.40[†] | 88.55[†] | 88.59[†] | - | 85.61[†] | - | 88.47[†] |

Table 2: Cosine learning rate scheduling train ImageNet. * is reported in Zhang et al. (2023)

| Model | ResNet18 | ResNet34 | ResNet50 |
|---|---|---|---|
| SGDF | **70.16** | **73.37** | **76.03** |
| SGD | 69.80 | 73.26 | 76.01* |

**Object Detection.** We conducted object detection experiments on the PASCAL VOC dataset (Everingham et al., 2010). The model used in these experiments was pre-trained on the COCO dataset (Lin et al., 2014), obtained from the official website. We trained this model on the VOC2007 and VOC2012 trainval dataset (17K) and evaluated it on the VOC2007 test dataset (5K). The utilized model was Faster-RCNN (Ren et al., 2015) with FPN, and the backbone was ResNet50 (He et al., 2016). Results are summarized in Tab. 3. To facilitate result reproduction, we provide the parameter table for this subpart in Tab. 7. As expected, SGDF outperforms other methods. These results also illustrate the efficiency of our method in object detection tasks.

Table 3: The mAP on PASCAL VOC using Faster-RCNN+FPN.

| Method | SGDF | SGD | Adam | AdamW | RAdam |
|---|---|---|---|---|---|
| mAP | **83.81** | 80.43 | 78.67 | 78.48 | 75.21 |

**Image Generation.** The stability of optimizers is crucial, especially when training Generative Adversarial Networks (GANs). If the generator and discriminator have mismatched complexities, it can lead to imbalance during GAN training, causing the GAN to fail to converge. This is known as model collapse. For instance, Vanilla SGD frequently causes model collapse, making adaptive optimizers like Adam and RMSProp the preferred choice. Therefore, GAN training provides a good benchmark for assessing optimizer stability. For reproducibility details, please refer to the parameter table in Tab. 8.

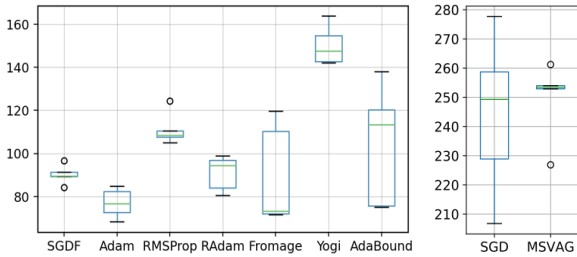

Figure 4: FID score of WGAN-GP.

We evaluated the Wasserstein-GAN with gradient penalty (WGAN-GP) (Salimans et al., 2016). Using well-known optimizers (Bernstein et al., 2020; Zaheer et al., 2018), the model was trained for 100 epochs. We then calculated the Frechet Inception Distance (FID) (Heusel et al., 2017) which is a metric that measures the similarity between the real image and the generated image distribution and is used to assess the quality of the generated model (lower FID indicates superior performance). Five random runs were conducted, and the outcomes are presented in Fig.4. Results for SGD and MSVAG were extracted from (Zhuang et al., 2020).

Experimental results demonstrate that SGDF significantly enhances WGAN-GP model training, achieving a FID score higher than vanilla SGD and outperforming most adaptive optimization methods. The integration of a Wiener filter in SGDF facilitates smooth gradient updates, mitigating training oscillations and effectively addressing the issue of pattern collapse.

## 4.2 TOP EIGENVALUES OF HESSIAN AND HESSIAN TRACE

The success of optimization algorithms in deep learning not only depends on their ability to minimize training loss, but also critically hinges on the nature of the solutions they converge to. We numerically verified the hessian matrix properties between the different methods.

We computed the Hessian spectrum of ResNet-18 trained on the CIFAR-100 dataset for 200 epochs using four optimization methods: SGD, SGDM, Adam, and SGDF. These experiments ensure that all methods achieve similar results on the training set. We employed power iteration (Yao et al., 2018) to compute the top eigenvalues of Hessian and Hutchinson's method (Yao et al., 2020a) to compute the Hessian trace. Histograms illustrating the distribution of the top 50 Hessian eigenvalues for each optimization method are presented in Fig. 5.

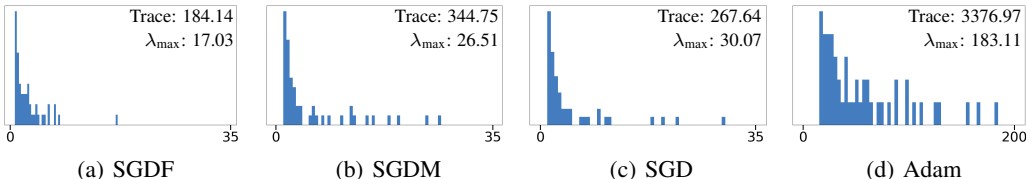

     (a) SGDF          (b) SGDM          (c) SGD          (d) Adam

Figure 5: Histogram of Top 50 Hessian Eigenvalues. The lower the value, the better the results of the test dataset.

### 4.3 VISUALIZATION OF LANDSCAPES

We visualized the loss landscapes of models trained with SGD, SGDM, SGDF, and Adam using the ResNet-18 model on CIFAR-100, following the method in (Li et al., 2018). All models are trained with the same hyperparameters for 200 epochs, as detailed in Sec. 4.1. As shown in Fig. 6, SGDF finds flatter minima. Notably, the visualization reveals that Adam is more prone to converge to sharper minima.

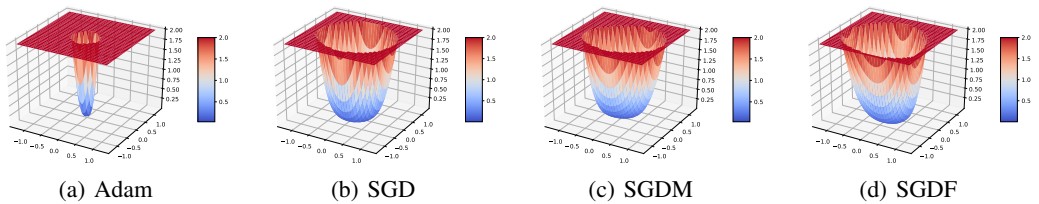

     (a) Adam          (b) SGD          (c) SGDM          (d) SGDF

Figure 6: Visualization of loss landscape. Adam converges to sharp minima.

### 4.4 WIENER FILTER COMBINES ADAM

We've conducted comparative experiments on the CIFAR-100 dataset, evaluating both the vanilla Adam algorithm and Wiener Adam, which substitutes the first-moment gradient estimates in the Adam optimizer with Wiener filter estimates. The results are presented in Tab. 4, and the detailed test curves are depicted in Fig. 11. This suggests that our first-moment filter estimation method has the potential to be applied to other optimization methods.

Table 4: Accuracy comparison between Adam and Wiener-Adam.

| Model | VGG11 | ResNet34 | DenseNet121 |
|---|---|---|---|
| Wiener-Adam | **62.64** | **73.98** | **74.89** |
| Vanilla-Adam | 56.73 | 72.34 | **74.89** |

For VGG without BN, the Wiener filter significantly improves performance by providing more accurate gradient estimates, reducing noise-induced errors, and ultimately enhancing accuracy. In contrast, for ResNet and DenseNet, which already incorporate BN and leverage residual and dense connections to stabilize gradient flow, the benefits of the Wiener filter are less pronounced. These architectures inherently promote stable gradient updates through their structural design, reducing the

additional advantages offered by the Wiener filter. This explains why the performance improvements vary across different architectures, as seen in Tab. 4. While Wiener-Adam provides a notable boost in simpler architectures like VGG, its impact is diminished in more complex networks where existing mechanisms already aid gradient stability.

## 5 RELATED WORKS

**Variance Reduction to Adaptive Methods.** In the early stages of deep learning development, optimization algorithms focused on reducing the variance of gradient estimation (Balles & Hennig, 2018; Defazio et al., 2014; Johnson & Zhang, 2013; Schmidt et al., 2017) to achieve a linear convergence rate. Subsequently, the emergence of adaptive learning rate methods (Dozat, 2016; Duchi et al., 2011; Zeiler, 2012) marked a significant shift in optimization algorithms. While SGD and its variants have advanced many applications, they come with inherent limitations. They often oscillate or become trapped in sharp minima (Wilson et al., 2017). Although these methods can lead models to achieve low training loss, such minima frequently fail to generalize effectively to new data (Hardt et al., 2015; Xie et al., 2022). This issue is exacerbated in the high-dimensional, non-convex landscapes characteristic of deep learning settings (Dauphin et al., 2014; Lucchi et al., 2022).

**Sharp and Flat Solutions.** The generalization ability of a deep learning model depends heavily on the nature of the solutions found during the optimization process. Keskar *et al.* (Keskar et al., 2017) demonstrated experimentally that flat minima generalize better than sharp minima. SAM (Foret et al., 2021) theoretically showed that the generalization error of smooth minima is lower than that of sharp minima on test data, and further proposed optimizing the zero-order smoothness. GAM (Zhang et al., 2023) improves SAM by simultaneously optimizing the prediction error and the number of paradigms of the maximum gradient in the neighborhood during the training process. Adaptive Inertia (Xie et al., 2020) aims to balance exploration and exploitation in the optimization process by adjusting the inertia of each parameter update. This adaptive inertia mechanism helps the model avoid falling into sharp local minima.

**Second-Order and Filter Methods.** The recent integration of second-order information into optimization problems has gained popularity (Liu et al., 2023; Yao et al., 2020b). Methods such as Kalman Filter (Kalman, 1960) combined with Gradient Descent incorporate second-order curvature information (Ollivier, 2019; Vuckovic, 2018). The KOALA algorithm (Davtyan et al., 2022) posits that the optimizer must adapt to the loss landscape. It adjusts learning rates based on both gradient magnitudes and the curvature of the loss landscape. However, it should be noted that the Kalman filtering framework introduces more complex parameter settings, which can hinder understanding and application.

## 6 CONCLUSION

In this paper, we introduce SGDF, a novel optimization method that estimates the gradient for faster convergence by leveraging both the variance of historical gradients and the current gradient. We demonstrate that SGDF yields solutions with a flat spectrum akin to SGD through Hessian spectral analysis. Through extensive experiments employing various deep learning architectures on benchmark datasets, we showcase SGDF's superior performance compared to other state-of-the-art optimizers, striking a balance between convergence speed and generalization.

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

## A  METHOD DERIVATION (SECTION 3 IN MAIN PAPER)

### A.1  WIENER FILTER DERIVATION FOR GRADIENT ESTIMATION (MAIN PAPER SECTION 3.1)

Given the sequence of gradients $\{g_t\}$ in a stochastic gradient descent process, we aim to find an estimate $\widehat{g}_t$ that incorporates information from both the historical gradients and the current gradient. The Wiener Filter provides an estimate that minimizes the mean squared error. We begin by constructing the estimate as a simple average and then refine it using the properties of the Wiener Filter.

$$
\begin{aligned}
\widehat{g}_t &= \frac{1}{T+1}\sum_{t=1}^{T} g_t + \frac{1}{T+1}g_t \\
&= \frac{1}{T+1}\frac{T}{T}\sum_{t=1}^{T} g_t + \frac{1}{T+1}g_t \\
&= \frac{T}{T+1}\bar{g}_t + \frac{1}{T+1}g_t \\
&\overset{(a)}{\approx} \frac{T}{T+1}\widehat{m}_t + \frac{1}{T+1}g_t \\
&= \left(1 - \frac{1}{T+1}\right)\widehat{m}_t + \frac{1}{T+1}g_t \\
&= \widehat{m}_t - K_t\widehat{m}_t + K_t g_t \\
&= \widehat{m}_t + K_t\left(g_t - \widehat{m}_t\right)
\end{aligned}
\tag{6}
$$

In the above derivation, step (a) replaces the arithmetic mean of gradients $\bar{g}_T$ with the momentum term $\widehat{m}_T$. The Wiener gain $K_T = \frac{1}{T+1}$ is then introduced to update the gradient estimate with information from the new gradient.

By defining $\widehat{g}_t$ as the weighted combination of the momentum term $\widehat{m}_t$ and the current gradient $g_t$, we can compute the variance of $\widehat{g}_t$ as follows:

$$
\begin{aligned}
\mathrm{Var}(\widehat{g}_t) &= \mathrm{Var}((1 - K_t)\widehat{m}_t + K_t g_t) \\
&= (1 - K_t)^2\mathrm{Var}(\widehat{m}_t) + K_t^2\mathrm{Var}(g_t)
\end{aligned}
\tag{7}
$$

Minimizing the variance of $\widehat{g}_t$ with respect to $K_t$, by setting the derivative $\dfrac{\mathrm{d}\mathrm{Var}(\widehat{g}_t)}{\mathrm{d}K_t} = 0$, yields:

$$
\begin{aligned}
0 &= 2(1 - K_t)\mathrm{Var}(\widehat{m}_t) + 2K_t\mathrm{Var}(g_t) \\
0 &= (1 - K_t)\mathrm{Var}(\widehat{m}_t) + K_t\mathrm{Var}(g_t) \\
K_t &= \frac{\mathrm{Var}(\widehat{m}_t)}{\mathrm{Var}(\widehat{m}_t) + \mathrm{Var}(g_t)}
\end{aligned}
\tag{8}
$$

The final expression for $K_t$ shows that the optimal interpolation coefficient is the ratio of the variance of the momentum term to the sum of the variances of the momentum term and the current gradient. This result exemplifies the essence of the Wiener Filter: optimally combining past information with new observations to reduce estimation error due to noisy data.

### A.2  VARIANCE CORRECTION (CORRECTION FACTOR IN MAIN PAPER SECTION 3.1)

The momentum term is defined as:

$$
m_t = (1 - \beta_1)\sum_{i=1}^{t}\beta_1^{t-i}g_{t-i+1},
\tag{9}
$$

which means that the momentum term is a weighted sum of past gradients, where the weights decrease exponentially over time.

To compute the variance of the momentum term $m_t$, we first observe that since $g_{t-i+1}$ are independent and identically distributed with a constant variance $\sigma_g^2$, the variance of the momentum term can be obtained by summing up the variances of all the weighted gradients.

The variance of each weighted gradient $\beta_1^{t-i} g_{t-i+1}$ is $\beta_1^{2(t-i)} \sigma_g^2$, because the variance operation has a quadratic nature, so the weight $\beta_1^{t-i}$ becomes $\beta_1^{2(t-i)}$ in the variance computation.

Therefore, the variance of $m_t$ is the sum of all these weighted variances:

$$\sigma_{m_t}^2 = (1 - \beta_1)^2 \sigma_g^2 \sum_{i=1}^{t} \beta_1^{2(t-i)}. \tag{10}$$

The factor $(1 - \beta_1)^2$ comes from the multiplication factor $(1 - \beta_1)$ in the momentum update formula, which is also squared when calculating the variance.

The summation part $\sum_{i=1}^{t} \beta_1^{2(t-i)}$ is a geometric series, which can be formulated as:

$$\sum_{i=1}^{t} \beta_1^{2(t-i)} = \frac{1 - \beta_1^{2t}}{1 - \beta_1^2}. \tag{11}$$

As $t \to \infty$, and given that $\beta_1 < 1$, we note that $\beta_1^{2t} \to 0$, and the geometric series sum converges to:

$$\sum_{i=1}^{t} \beta_1^{2(t-i)} = \frac{1 - \beta_1^{2t}}{1 - \beta_1^2} = \frac{1}{1 - \beta_1^2}. \tag{12}$$

Consequently, the long-term variance of the momentum term $m_t$ is expressed as:

$$\sigma_{m_t}^2 = \left( \frac{1 - \beta_1}{1 - \beta_1^2} \right)^2 \sigma_g^2 = \frac{1 - \beta_1}{1 + \beta_1} \sigma_g^2. \tag{13}$$

This result shows how the effective gradient noise is reduced by the momentum term, which is a factor of $\frac{1-\beta_1}{1+\beta_1}$ compared to the variance of the gradients $\sigma_g^2$.

### A.3 FUSION GAUSSIAN DISTRIBUTION (MAIN PAPER SECTION 3.2)

Consider two Gaussian distributions for the momentum term $\widehat{m}_t$ and the current gradient $g_t$:

- The momentum term $\widehat{m}_t$ is normally distributed with mean $\mu_m$ and variance $\sigma_m^2$, denoted as $\widehat{m}_t \sim \mathcal{N}(\mu_m, \sigma_m^2)$.
- The current gradient $g_t$ is normally distributed with mean $\mu_g$ and variance $\sigma_g^2$, denoted as $g_t \sim \mathcal{N}(\mu_g, \sigma_g^2)$.

The product of their probability density functions is given by:

$$N(\widehat{m}_t; \mu_m, \sigma_m) \cdot N(g_t; \mu_g, \sigma_g) = \frac{1}{2\pi\sigma_m\sigma_g} \exp\left( -\frac{(\widehat{m}_t - \mu_m)^2}{2\sigma_m^2} - \frac{(g_t - \mu_g)^2}{2\sigma_g^2} \right) \tag{14}$$

The goal is to find equivalent mean $\mu'$ and variance $\sigma'^2$ for the new Gaussian distribution that matches the product:

$$N(x; \mu', \sigma'^2) = \frac{1}{\sqrt{2\pi}\sigma'} \exp\left( -\frac{(x - \mu')^2}{2\sigma'^2} \right) \tag{15}$$

We derive the expression for combining these two distributions. For convenience, let us define the variable $t$ as follows:

$$
\begin{aligned}
t &= -\frac{(x - \mu_m)^2}{2\sigma_m^2} - \frac{(x - \mu_g)^2}{2\sigma_g^2} \\
&= -\frac{\sigma_g^2 (x - \mu_m)^2 + \sigma_m^2 (x - \mu_g)^2}{2\sigma_m^2 \sigma_g^2} \\
&= -\frac{\left( x - \frac{\sigma_g^2 \mu_m + \sigma_m^2 \mu_g}{\sigma_m^2 + \sigma_g^2} \right)^2}{\frac{2\sigma_m^2 \sigma_g^2}{\sigma_m^2 + \sigma_g^2}} + \frac{(\mu_m - \mu_g)^2}{2 (\sigma_m^2 + \sigma_g^2)}.
\end{aligned}
\tag{16}
$$

Through coefficient matching in the exponential terms, we obtain the new mean and variance:

$$\mu' = \frac{\sigma_g^2 \mu_m + \sigma_m^2 \mu_g}{\sigma_m^2 + \sigma_g^2} \quad \sigma'^2 = \frac{\sigma_m^2 \sigma_g^2}{\sigma_m^2 + \sigma_g^2} \tag{17}$$

The new mean $\mu'$ is a weighted average of the two means, $\mu_m$ and $\mu_g$, with weights inversely proportional to their variances. This places $\mu'$ between $\mu_m$ and $\mu_g$, closer to the mean with the smaller variance. The new standard deviation $\sigma'$ is smaller than either of the original standard deviations $\sigma_m$ and $\sigma_g$, which reflects the reduced uncertainty in the estimate due to the combination of information from both sources. This is a direct consequence of the Wiener Filter's optimality in the mean-square error sense.

### A.4 FOKKER PLANCK MODELLING (THEOREM 3.1 IN MAIN PAPER)

**Theorem A.1.** *Consider a system described by the Fokker-Planck equation, evolving the probability density function $P$ in one-dimensional and multi-dimensional parameter spaces. Given a loss function $f(\theta)$, and the noise variance $D$ or diffusion matrix $D_{ij}$ satisfying $D \geq C > 0$ or $D_i \geq C > 0$, where $C$ is a positive lower bound constant, known as the Cramér-Rao lower bound. In the steady state condition, i.e., $\frac{\partial P}{\partial t} = 0$, the analytical form of the probability density $P$ can be obtained by solving the corresponding Fokker-Planck equation. These solutions reveal the probability distribution of the system at steady state, described as follows:*

***One-dimensional case*** *In a one-dimensional parameter space, the probability density function $P(\theta)$ is*

$$P(\theta) = \frac{1}{Z} \exp\left(-\int \frac{1}{D}\frac{\partial f}{\partial \theta} dx\right), \tag{18}$$

*where $Z$ is a normalization constant, ensuring the total probability sums to one.*

***Multi-dimensional case*** *In a multi-dimensional parameter space, the probability density function $P(\theta)$ is*

$$P(\theta) = \frac{1}{Z} \exp\left(-\sum_{i=1}^{n} \frac{f(\theta)}{D_i}\right), \tag{19}$$

*Here, $Z$ is also a normalization constant, ensuring the total probability sums to one, assuming $D_{ij} = D_i \delta_{ij}$, where $\delta_{ij}$ is the Kronecker delta.*

**Proof.**

**one-dimensional Fokker-Planck equation:** Given the one-dimensional Fokker-Planck equation:

$$\frac{\partial P}{\partial t} = -\frac{\partial}{\partial \theta}\left(P\frac{\partial f}{\partial \theta}\right) + \frac{\partial^2}{\partial \theta^2}(DP), \tag{20}$$

where $f(\theta)$ is the loss function, and $D$ is the variance of the noise, with $D \geq C > 0$ representing a positive lower bound for the variance. $P$ denotes the probability density of finding the state of the system near a given point or region

**Derivation of the Steady-State Distribution:**

In the steady state condition, $\frac{\partial P}{\partial t} = 0$, thus the equation simplifies to:

$$0 = -\frac{\partial}{\partial \theta}\left(P\frac{\partial f}{\partial \theta}\right) + \frac{\partial^2}{\partial \theta^2}(DP). \tag{21}$$

Our goal is to find the probability density $P$ as a function of $\theta$.

By integrating, we obtain:

$$\frac{\partial}{\partial \theta}\left(P\frac{\partial f}{\partial \theta}\right) = \frac{\partial^2}{\partial \theta^2}(DP). \tag{22}$$

Next, we set $J = P\frac{\partial f}{\partial \theta}$ as the probability current, and we have:

$$\frac{\partial J}{\partial \theta} = \frac{\partial}{\partial \theta}\left(D\frac{\partial P}{\partial \theta}\right). \tag{23}$$

Upon integration, we get:

$$J = D\frac{\partial P}{\partial \theta} + C_1, \tag{24}$$

where $C_1$ is an integration constant. Assuming the probability current $J$ vanishes at infinity, then $C_1 = 0$.

Therefore, we have:

$$D\frac{\partial P}{\partial \theta} = P\frac{\partial f}{\partial \theta}. \tag{25}$$

This equation can be rewritten as:

$$\frac{\partial P}{\partial \theta} = \frac{P}{D}\frac{\partial f}{\partial \theta}. \tag{26}$$

Now, leveraging the variance lower bound $D \geq C$, we analyze the above equation. Since $D$ is a positive constant, we can further integrate to get $P$:

$$\ln P = -\int \frac{1}{D}\frac{\partial f}{\partial \theta}d\theta + C_2, \tag{27}$$

where $C_2$ is an integration constant.

Solving for $P$, we get:

$$P = e^{C_2}\exp\left(-\int \frac{1}{D}\frac{\partial f}{\partial \theta}d\theta\right). \tag{28}$$

Since we know that $D$ has a lower bound, $\frac{1}{D}$ is bounded above, which suggests that $P$ will not explode at any specific value of $\theta$.

**multi-dimensional Fokker-Planck equation:** Consider a multi-dimensional parameter space $x \in \mathbb{R}^n$ and a loss function $f(\theta)$. The evolution of the probability density function $P(\theta, t)$ in this space governed by the Fokker-Planck equation is given by:

$$\frac{\partial P}{\partial t} = -\sum_{i=1}^{n}\frac{\partial}{\partial \theta_i}\left(P\frac{\partial f}{\partial \theta_i}\right) + \sum_{i=1}^{n}\sum_{j=1}^{n}\frac{\partial^2}{\partial \theta_i \partial \theta_j}(D_{ij}P), \tag{29}$$

where $D_{ij}$ are elements of the diffusion matrix, representing the intensity and correlation of the stochastic in the directions $\theta_i$ and $\theta_j$. At the steady state, where the time derivative of $P$ vanishes, we find:

$$0 = -\sum_{i=1}^{n}\frac{\partial}{\partial \theta_i}\left(P\frac{\partial f}{\partial \theta_i}\right) + \sum_{i=1}^{n}\sum_{j=1}^{n}\frac{\partial^2}{\partial \theta_i \partial \theta_j}(D_{ij}P). \tag{30}$$

Assuming $D_{ij} = D_i\delta_{ij}$ where $\delta_{ij}$ is the Kronecker delta, and $D_i \geq C > 0$, the equation simplifies to:

$$0 = -\sum_{i=1}^{n}\frac{\partial}{\partial \theta_i}\left(P\frac{\partial f}{\partial \theta_i}\right) + \sum_{i=1}^{n}\frac{\partial^2}{\partial \theta_i^2}(D_iP). \tag{31}$$

Integrating with respect to $\theta_i$, we obtain a set of equations:

$$D_i\frac{\partial P}{\partial \theta_i} = P\frac{\partial f}{\partial \theta_i} + C_i, \tag{32}$$

where $C_i$ is an integration constant. Assuming $C_i = 0$, which corresponds to no flux at the boundaries, we can solve for $P$:

$$P(\theta) = \frac{1}{Z} \exp\left( -\sum_{i=1}^{n} \frac{f(\theta)}{D_i} \right), \tag{33}$$

where $Z$ is a normalization constant ensuring that the total probability integrates to one.

**Exploration Efficacy of SGD due to Variance Lower Bound** The existence of a variance lower bound in Stochastic Gradient Descent (SGD) critically enhances the algorithm's exploration capabilities, particularly in regions of the loss landscape where gradients are minimal. By preventing the probability density function from becoming unbounded, it ensures continuous exploration and increases the probability of converging to flat minima that are associated with better generalization properties. This principle holds true across both one-dimensional and multi-dimensional scenarios, making the variance lower bound an essential consideration for optimizing SGD's performance in finding robust, generalizable solutions.

# B CONVERGENCE ANALYSIS IN CONVEX ONLINE LEARNING CASE (THEOREM 3.2 IN MAIN PAPER).

**Assumption B.1.** Variables are bounded: $\exists D$ such that $\forall t, \|\theta_t\|_2 \leq D$. Gradients are bounded: $\exists G$ such that $\forall t, \|g_t\|_2 \leq G$.

**Definition B.2.** Let $f_t(\theta_t)$ be the loss at time $t$ and $f_t(\theta^*)$ be the loss of the best possible strategy at the same time. The cumulative regret $R(T)$ at time $T$ is defined as:

$$R(T) = \sum_{t=1}^{T} f_t(\theta_t) - f_t(\theta^*) \tag{34}$$

**Definition B.3.** If a function $f \colon R^d \to R$ is convex if for all $x, y \in R^d$ for all $\lambda \in [0, 1]$,

$$\lambda f(x) + (1 - \lambda) f(y) \geq f(\lambda x + (1 - \lambda) y) \tag{35}$$

Also, notice that a convex function can be lower bounded by a hyperplane at its tangent.

**Lemma B.4.** *If a function* $f : R^d \to R$ *is convex, then for all* $x, y \in R^d$ ,

$$f(y) \geq f(x) + \nabla f(x)^T (y - x) \tag{36}$$

The above lemma can be used to upper bound the regret, and our proof for the main theorem is constructed by substituting the hyperplane with SGDF update rules.

The following two lemmas are used to support our main theorem. We also use some definitions to simplify our notation, where $g_t \triangleq \nabla f_t(\theta_t)$ and $g_{t,i}$ as the $i^{\text{th}}$ element. We denote $g_{1:t,i} \in \mathbb{R}^t$ as a vector that contains the $i^{\text{th}}$ dimension of the gradients over all iterations till $t$, $g_{1:t,i} = [g_{1,i}, g_{2,i}, \cdots, g_{t,i}]$

**Lemma B.5.** *Let* $g_t = \nabla f_t(\theta_t)$ *and* $g_{1:t}$ *be defined as above and bounded,*

$$\|g_t\|_2 \leq G, \|g_t\|_\infty \leq G_\infty. \tag{37}$$

*Then,*

$$\sum_{t=1}^{T} g_{t,i} \leq 2 G_\infty \|g_{1:T,i}\|_2. \tag{38}$$

*Proof.* We will prove the inequality using induction over $T$. For the base case $T = 1$:

$$g_{1,i} \leq 2 G_\infty \|g_{1,i}\|_2. \tag{39}$$

Assuming the inductive hypothesis holds for $T - 1$, for the inductive step:

$$\begin{aligned} \sum_{t=1}^{T} g_{t,i} &= \sum_{t=1}^{T-1} g_{t,i} + g_{T,i} \\ &\leq 2 G_\infty \|g_{1:T-1,i}\|_2 + g_{T,i} \\ &= 2 G_\infty \sqrt{\|g_{1:T,i}\|_2^2 - g_T^2} + g_{T,i}^2. \end{aligned} \tag{40}$$

Given,

$$\|g_{1:T,i}\|_2^2 - g_{T,i}^2 + \frac{g_{T,i}^4}{4 \|g_{1:T,i}\|_2^2} \geq \|g_{1:T,i}\|_2^2 - g_{T,i}^2, \tag{41}$$

taking the square root of both sides, we get:

$$\begin{aligned} \sqrt{\|g_{1:T,i}\|_2^2 - g_{T,i}^2} &\leq \|g_{1:T,i}\|_2 - \frac{g_{T,i}^2}{2 \|g_{1:T,i}\|_2} \\ &\leq \|g_{1:T,i}\|_2 - \frac{g_{T,i}^2}{2 \sqrt{G_\infty^2}}. \end{aligned} \tag{42}$$

Substituting into the previous inequality:

$$G_\infty \sqrt{\|g_{1:T,i}\|_2^2 - g_{T,i}^2} + \sqrt{g_{T,i}^2} \le 2G_\infty \|g_{1:T,i}\|_2 \tag{43}$$

**Lemma B.6.** *Let bounded $g_t$, $\|g_t\|_2 \le G$, $\|g_t\|_\infty \le G_\infty$, the following inequality holds*

$$\sum_{t=1}^T \widehat{m}_{t,i}^2 \le \frac{4G_\infty^2}{(1-\beta_1)^2} \|g_{1:T,i}\|_2^2 \tag{44}$$

**Proof.** Under the inequality: $\frac{1}{(1-\beta_1^t)^2} \le \frac{1}{(1-\beta_1)^2}$ . We can expand the last term in the summation using the updated rules in Algorithm 1,

$$
\begin{aligned}
\sum_{t=1}^T \widehat{m}_{t,i}^2 &= \sum_{t=1}^{T-1} \widehat{m}_{t,i}^2 + \frac{\left(\sum_{k=1}^T (1-\beta_1)\beta_1^{T-k} g_{k,i}\right)^2}{\left(1-\beta_1^T\right)^2} \\
&\le \sum_{t=1}^{T-1} \widehat{m}_{t,i}^2 + \frac{\sum_{k=1}^T T\left((1-\beta_1)\beta_1^{T-k} g_{k,i}\right)^2}{\left(1-\beta_1^T\right)^2} \\
&\le \sum_{t=1}^{T-1} \widehat{m}_{t,i}^2 + \frac{(1-\beta_1)^2}{\left(1-\beta_1^T\right)^2} \sum_{k=1}^T T\left(\beta_1^2\right)^{T-k} \|g_{k,i}\|_2^2 \\
&\le \sum_{t=1}^{T-1} \widehat{m}_{t,i}^2 + T \sum_{k=1}^T \left(\beta_1^2\right)^{T-k} \|g_{k,i}\|_2^2
\end{aligned}
\tag{45}
$$

Similarly, we can upper-bound the rest of the terms in the summation.

$$
\begin{aligned}
\sum_{t=1}^T \widehat{m}_{t,i}^2 &\le \sum_{t=1}^T \|g_{t,i}\|_2^2 \sum_{j=0}^{T-t} t\beta_1^j \\
&\le \sum_{t=1}^T \|g_{t,i}\|_2^2 \sum_{j=0}^T t\beta_1^j
\end{aligned}
\tag{46}
$$

For $\beta_1 < 1$ , using the upper bound on the arithmetic-geometric series, $\sum_t t\beta_1^t < \frac{1}{(1-\beta_1)^2}$ :

$$\sum_{t=1}^T \|g_{t,i}\|_2^2 \sum_{j=0}^T t\beta_1^j \le \frac{1}{(1-\beta_1)^2} \sum_{t=1}^T \|g_{t,i}\|_2^2 \tag{47}$$

Apply Lemma B.5,

$$\sum_{t=1}^T \widehat{m}_{t,i}^2 \le \frac{4G_\infty^2}{(1-\beta_1)^2} \|g_{1:T,i}\|_2^2 \tag{48}$$

**Theorem B.7.** *Assume that the function $f_t$ has bounded gradients, $\|\nabla f_t(\theta)\|_2 \le G$, $\|\nabla f_t(\theta)\|_\infty \le G_\infty$ for all $\theta \in \mathbb{R}^d$ and the distance between any $\theta_t$ generated by SGDF is bounded, $\|\theta_n - \theta_m\|_2 \le D$, $\|\theta_m - \theta_n\|_\infty \le D_\infty$ for any $m,n \in \{1,...,T\}$, and $\beta_1, \beta_2 \in [0,1)$. Let $\alpha_t = \alpha/\sqrt{t}$. For all $T \ge 1$, SGDF achieves the following guarantee:*

$$R(T) \le \frac{D^2}{\alpha} \sum_{i=1}^d \sqrt{T} + \frac{2D_\infty G_\infty}{1-\beta_1} \sum_{i=1}^d \|g_{1:T,i}\|_2 + \frac{2\alpha G_\infty^2 (1+(1-\beta_1)^2)}{\sqrt{T}(1-\beta_1)^2} \sum_{i=1}^d \|g_{1:T,i}\|_2^2 \tag{49}$$

**Proof of convex Convergence.**

We aim to prove the convergence of the algorithm by showing that $R(T)$ is bounded, or equivalently, that $\frac{R(T)}{T}$ converges to zero as $T$ goes to infinity.

To express the cumulative regret in terms of each dimension, let $f_t(\theta_t)$ and $f_t(\theta^*)$ represent the loss and the best strategy's loss for the $d$th dimension, respectively. Define $R_{T,d}$ as:

$$R_{T,i} = \sum_{t=1}^{T} f_t(\theta_t) - f_t(\theta^*) \tag{50}$$

Then, the overall regret $R(T)$ can be expressed in terms of all dimensions $D$ as:

$$R(T) = \sum_{d=1}^{D} R_{T,i} \tag{51}$$

Establishing the Connection: From the Iteration of $\theta_t$ to $\langle g_t, \theta_t - \theta^* \rangle$

Using Lemma B.4, we have,

$$f_t(\theta_t) - f_t(\theta^*) \le g_t^T(\theta_t - \theta^*) = \sum_{i=1}^{d} g_{t,i}(\theta_{t,i} - \theta_{,i}^*) \tag{52}$$

From the update rules presented in algorithm 1,

$$\begin{aligned} \theta_{t+1} &= \theta_t - \alpha_t \widehat{g}_t \\ &= \theta_t - \alpha_t(\widehat{m}_t + K_{t,d}(g_t - \widehat{m}_t)) \end{aligned} \tag{53}$$

We focus on the $i^{\text{th}}$ dimension of the parameter vector $\theta_t \in R^d$. Subtract the scalar $\theta_{,i}^*$ and square both sides of the above update rule, we have,

$$(\theta_{t+1,d} - \theta_{,i}^*)^2 = (\theta_{t,i} - \theta_{,i}^*)^2 - 2\alpha_t(\widehat{m}_{t,i} + K_{t,d}(g_{t,i} - \widehat{m}_{t,i}))(\theta_{t,i} - \theta_{,i}^*) + \alpha_t^2 \widehat{g}_t^2 \tag{54}$$

Separating items $g_{t,i}(\theta_{t,i} - \theta_{,i}^*)$:

$$g_{t,d}(\theta_{t,i} - \theta_{,i}^*) = \underbrace{\frac{(\theta_{t,i} - \theta_{,i}^*)^2 - (\theta_{t+1,i} - \theta_{,i}^*)^2}{2\alpha_t K_{t,i}}}_{(1)} - \underbrace{\frac{1 - K_{t,i}}{K_{t,i}}\widehat{m}_{t,i}(\theta_{t,i} - \theta_{,i}^*)}_{(2)} + \underbrace{\frac{\alpha_t}{2K_{t,i}}(\widehat{g}_{t,i})^2}_{(3)} \tag{55}$$

We then deal with (1), (2) and (3) separately.

For the first term (1), we have:

$$\begin{aligned} &\sum_{t=1}^{T} \frac{(\theta_{t,i} - \theta_{,i}^*)^2 - (\theta_{t+1,i} - \theta_{,i}^*)^2}{2\alpha_t K_{t,i}} \\ &\le \sum_{t=1}^{T} \frac{(\theta_{t,i} - \theta_{,i}^*)^2 - (\theta_{t+1,i} - \theta_{,i}^*)^2}{2\alpha_t K_{t,i}} \\ &= \frac{(\theta_{1,i} - \theta_{,i}^*)^2}{2\alpha_1 K_{1,i}} - \frac{(\theta_{T+1,i} - \theta_{,i}^*)^2}{2\alpha_T K_{T,i}} + \sum_{t=2}^{T}(\theta_{t,i} - \theta_{,i}^*)^2 \left[ \frac{1}{2\alpha_t K_{t,i}} - \frac{1}{2\alpha_{t-1}K_{t-1,i}} \right] \end{aligned} \tag{56}$$

Given that $-\dfrac{(\theta_{T+1,i} - \theta_{,i}^*)^2}{2\alpha_T(K_1)} \le 0$ and $\dfrac{(\theta_{1,i} - \theta_{,i}^*)^2}{2\alpha_1(K_T)} \le \dfrac{D_i^2}{2\alpha_1(K_T)}$, we can bound it as:

$$\begin{aligned} &\sum_{t=1}^{T} \frac{(\theta_{t,i} - \theta_{,i}^*)^2 - (\theta_{t+1,i} - \theta_{,i}^*)^2}{2\alpha_t K_{t,i}} \\ &\le \sum_{i=1}^{d} \frac{(\theta_{t,i} - \theta_{,i}^*)^2}{2\alpha_t K_{t,i}} \end{aligned} \tag{57}$$

For the second term (2), we have:

$$
\sum_{t=1}^{T} -\frac{1-K_{t,i}}{K_{t,i}} \widehat{m}_{t,i} \left(\theta_{t,i} - \theta_{,i}^*\right)
$$

$$
= \sum_{t=1}^{T} -\frac{1-K_{t,i}}{K_{t,i}(1-\beta_1^t)} \left(\sum_{i=1}^{T}(1-\beta_{1,i}) \prod_{j=i+1}^{T} \beta_{1,j}\right) g_{t,i} \left(\theta_{t,i} - \theta_{,i}^*\right)
$$

$$
\leq \sum_{t=1}^{T} -\frac{1-K_{t,i}}{K_{t,d}(1-\beta_1^t)} \left(1 - \prod_{i=1}^{T} \beta_{1,i}\right) g_{t,i}(\theta_{t,i} - \theta_{,i}^*)
$$

$$
\leq \sum_{t=1}^{T} \frac{1-K_{t,i}}{K_{t,d}(1-\beta_1^t)} g_{t,i}(\theta_{t,i} - \theta_{,i}^*)
$$

(58)

For the third term (3), we have:

$$
\sum_{t=1}^{T} \frac{\alpha_t}{2K_{t,i}} (\widehat{g}_{t,i})^2 \leq \sum_{t=1}^{T} \frac{\alpha_t}{2K_{t,i}} \left(\widehat{m}_{t,i} + K_t(g_{t,i} - \widehat{m}_{t,i})\right)^2
$$

$$
\leq \sum_{t=1}^{T} \frac{\alpha_t}{2K_{t,i}} \left((1-K_{t,i})\widehat{m}_{t,i} + K_{t,d}g_{t,i}\right)^2
$$

$$
\leq \sum_{t=1}^{T} \frac{\alpha_t}{2K_{t,i}} \left(2(1-K_{t,i})^2 \widehat{m}_{t,i}^2 + 2K_{t,i}^2 g_{t,i}^2\right)
$$

$$
\leq \sum_{t=1}^{T} \frac{\alpha_t}{K_{t,i}} \left((1-K_{t,i})^2 \widehat{m}_{t,i}^2 + K_{t,i}^2 g_{t,i}^2\right)
$$

(59)

Collate all the items that we have:

$$
R(T) \leq \sum_{i=1}^{d}\sum_{t=1}^{T} \frac{(\theta_{t,i} - \theta_{,i}^*)^2}{2\alpha_t K_{t,i}} + \sum_{i=1}^{d}\sum_{t=1}^{T} \frac{1-K_{t,i}}{K_{t,i}(1-\beta_1^t)} g_{t,i}(\theta_{t,i} - \theta_{,i}^*) + \sum_{i=1}^{d}\sum_{t=1}^{T} \frac{\alpha_t}{K_{t,i}} \left((1-K_{t,i})^2 \widehat{m}_{t,i}^2 + K_{t,i}^2 g_{t,i}^2\right)
$$

(60)

Using Lemma B.5 and Lemma B.6 From $\sum_{t=1}^{T} \widehat{s}_t > \sum_{t=1}^{T}(g_t - \widehat{m}_t)^2$, we have $\frac{1}{T}\sum_{t=1}^{T} K_t > \frac{1}{2}$. Therefore, from the assumption, $\|\theta_t - \theta^*\|_2^2 \leq D, \|\theta_m - \theta_n\|_\infty \leq D_\infty$, we have the following regret bound:

$$
R(T) \leq \frac{D^2}{\alpha} \sum_{i=1}^{d} \sqrt{T} + \frac{2D_\infty G_\infty}{1-\beta_1} \sum_{i=1}^{d} \|g_{1:T,i}\|_2 + \frac{2\alpha G_\infty^2(1 + (1-\beta_1)^2)}{\sqrt{T}(1-\beta_1)^2} \sum_{i=1}^{d} \|g_{1:T,i}\|_2^2 \quad (61)
$$

# C  CONVERGENCE ANALYSIS FOR NON-CONVEX STOCHASTIC OPTIMIZATION (THEOREM 3.3 IN MAIN PAPER).

We have relaxed the assumption on the objective function, allowing it to be non-convex, and adjusted the criterion for convergence from the statistic $R(T)$ to $\mathbb{E}(T)$. Let's briefly review the assumptions and the criterion for convergence after relaxing the assumption:

**Assumption C.1.**

- A1 Bounded variables (same as convex). $\|\theta - \theta^*\|_2 \leq D, \ \forall \theta, \theta^*$ or for any dimension $i$ of the variable, $\|\theta_i - \theta_i^*\|_2 \leq D_i, \ \forall \theta_i, \theta_i^*$

- A2 The noisy gradient is unbiased. For $\forall t$, the random variable $\zeta_t$ is defined as $\zeta_t = g_t - \nabla f(\theta_t)$, $\zeta_t$ satisfy $\mathbb{E}[\zeta_t] = 0$, $\mathbb{E}\left[\|\zeta_t\|_2^2\right] \leq \sigma^2$, and when $t_1 \neq t_2$, $\zeta_{t_1}$ and $\zeta_{t_2}$ are statistically independent, i.e., $\zeta_{t_1} \perp \zeta_{t_2}$.

- A3 Bounded gradient and noisy gradient. At step $t$, the algorithm can access a bounded noisy gradient, and the true gradient is also bounded. *i.e.* $\|\nabla f(\theta_t)\| \leq G, \ \|g_t\| \leq G, \ \forall t > 1$.

- A4 The property of function. The objective function $f(\theta)$ is a global loss function, defined as $f(\theta) = \lim_{T \to \infty} \frac{1}{T} \sum_{t=1}^T f_t(\theta)$. Although $f(\theta)$ is no longer a convex function, it must still be a $L$-smooth function, i.e., it satisfies (1) $f$ is differentiable, $\nabla f$ exists everywhere in the domain; (2) there exists $L > 0$ such that for any $\theta_1$ and $\theta_2$ in the domain, (first definition)

$$f(\theta_2) \leq f(\theta_1) + \langle \nabla f(\theta_1), \theta_2 - \theta_1 \rangle + \frac{L}{2}\|\theta_2 - \theta_1\|_2^2 \tag{62}$$

  or (second definition)

$$\|\nabla f(\theta_1) - \nabla f(\theta_2)\|_2 \leq L\|\theta_1 - \theta_2\|_2 \tag{63}$$

  This condition is also known as *L - Lipschitz*.

**Definition C.2.** The criterion for convergence is the statistic $\mathbb{E}(T)$:

$$\mathbb{E}(T) = \min_{t=1,2,\ldots,T} \mathbb{E}_{t-1}\left[\|\nabla f(\theta_t)\|_2^2\right] \tag{64}$$

When $T \to \infty$, if the amortized value of $\mathbb{E}(T)$, $\mathbb{E}(T)/T \to 0$, we consider such an algorithm to be convergent, and generally, the slower $\mathbb{E}(T)$ grows with $T$, the faster the algorithm converges.

**Definition C.3.** Define $\xi_t$ as

$$\xi_t = \begin{cases} \theta_t & t = 1 \\ \theta_t + \frac{\beta_1}{1-\beta_1}(\theta_t - \theta_{t-1}) & t \geq 2 \end{cases} \tag{65}$$

**Theorem C.4.** *Consider a non-convex optimization problem. Suppose assumptions A1-A5 are satisfied, and let $\alpha_t = \alpha/\sqrt{t}$. For all $T \geq 1$, SGDF achieves the following guarantee:*

$$\mathbb{E}(T) \leq \frac{C_7 \alpha^2 (\log T + 1) + C_8}{2\alpha\sqrt{T}} \tag{66}$$

*where $\mathbb{E}(T) = \min_{t=1,2,\ldots,T} \mathbb{E}_{t-1}\left[\|\nabla f(\theta_t)\|_2^2\right]$ denotes the minimum of the squared-paradigm expectation of the gradient, $\alpha$ is the learning rate at the 1-th step, $C_7$ are constants independent of $d$ and $T$, $C_8$ is a constant independent of $T$, and the expectation is taken w.r.t all randomness corresponding to $g_t$.*

**Proof of convex Convergence.**

Since $f$ is an L-smooth function,

$$\|\nabla f(\xi_t) - \nabla f(\theta_t)\|_2^2 \leq L^2 \|\xi_t - \theta_t\|_2^2 \tag{67}$$

Thus,

$$f\left(\xi_{t+1}\right) - f\left(\xi_t\right)$$

$$\leq \langle \nabla f\left(\xi_t\right), \xi_{t+1} - \xi_t \rangle + \frac{L}{2} \left\| \xi_{t+1} - \xi_t \right\|_2^2$$

$$= \left\langle \frac{1}{\sqrt{L}} \left(\nabla f\left(\xi_t\right) - \nabla f\left(\theta_t\right)\right), \sqrt{L}\left(\xi_{t+1} - \xi_t\right) \right\rangle + \langle \nabla f\left(\theta_t\right), \xi_{t+1} - \xi_t \rangle + \frac{L}{2} \left\| \xi_{t+1} - \xi_t \right\|_2^2$$

$$\leq \frac{1}{2} \left( \frac{1}{L} \left\| \nabla f\left(\xi_t\right) - \nabla f\left(\theta_t\right) \right\|_2^2 + L \left\| \xi_{t+1} - \xi_t \right\|_2^2 \right) + \langle \nabla f\left(\theta_t\right), \xi_{t+1} - \xi_t \rangle + \frac{L}{2} \left\| \xi_{t+1} - \xi_t \right\|_2^2$$

$$\leq \frac{1}{2L} \left\| \nabla f\left(\xi_t\right) - \nabla f\left(\theta_t\right) \right\|_2^2 + L \left\| \xi_{t+1} - \xi_t \right\|_2^2 + \langle \nabla f\left(\theta_t\right), \xi_{t+1} - \xi_t \rangle$$

$$\leq \frac{1}{2L} L^2 \left\| \xi_t - \theta_t \right\|_2^2 + L \left\| \xi_{t+1} - \xi_t \right\|_2^2 + \langle \nabla f\left(\theta_t\right), \xi_{t+1} - \xi_t \rangle$$

$$= \frac{L}{2} \underbrace{\left\| \xi_t - \theta_t \right\|_2^2}_{(1)} + L \underbrace{\left\| \xi_{t+1} - \xi_t \right\|_2^2}_{(2)} + \underbrace{\langle \nabla f\left(\theta_t\right), \xi_{t+1} - \xi_t \rangle}_{(3)}$$

$$(68)$$

Next, we will deal with the three terms (1), (2), and (3) separately.

**For term (1)**

When $t = 1$, $\left\| \xi_t - \theta_t \right\|_2^2 = 0$

When $t \geq 2$,

$$\begin{aligned}
\left\| \xi_t - \theta_t \right\|_2^2 &= \left\| \frac{\beta_1}{1 - \beta_1} \left(\theta_t - \theta_{t-1}\right) \right\|_2^2 \\
&= \frac{\beta_1^2}{\left(1 - \beta_1\right)^2} \alpha_{t-1}^2 \left\| \widehat{g}_{t-1,i} \right\|_2^2 \\
&= \frac{\beta_1^2}{\left(1 - \beta_1\right)^2} \alpha_{t-1}^2 \sum_{i=1}^d \left(1 - K_t\right) \left(\widehat{m}_{t-1,i}\right)^2 + K_t g_t^2 \\
&\overset{(a)}{\leq} \frac{\beta_1^2}{\left(1 - \beta_1\right)^2} \alpha_{t-1}^2 \sum_{i=1}^d G_i^2
\end{aligned}$$

$$(69)$$

Where (a) holds because for any $t$:

- $\left| \widehat{m}_{t,i} \right| \leq \frac{1}{1 - \beta_1^t} \sum_{s=1}^t \left(1 - \beta_1\right) \beta_1^{t-s} \left| g_{s,i} \right| \leq \frac{1}{1 - \beta_1^t} \sum_{s=1}^t \left(1 - \beta_1\right) \beta_1^{t-s} G_i = G_i$.

- $\left\| g_t \right\|_2 \leq G$, $\forall t$, or for any dimension of the variable $i$: $\left\| g_{t,i} \right\|_2 \leq G_i$, $\forall t$

**For term (2)**

When $t = 1$,

$$
\begin{aligned}
\xi_{t+1} - \xi_t &= \theta_{t+1} + \frac{\beta_1}{1 - \beta_1} \left(\theta_{t+1} - \theta_t\right) - \theta_t \\
&= \frac{1}{1 - \beta_1} \left(\theta_{t+1} - \theta_t\right) \\
&= -\frac{\alpha_t}{1 - \beta_1} \left(\widehat{g}_t\right) \\
&= -\frac{\alpha_t}{1 - \beta_1} \left(\frac{1 - K_t}{1 - \beta_1^t} m_t + K_t g_t\right) \\
&= -\frac{\alpha_t}{1 - \beta_1} \frac{1 - K_t}{1 - \beta_1^t} \left(\beta_1 \underset{\nearrow\,0}{m_{t-1}} + (1 - \beta_1) g_t\right) - \frac{\alpha_t}{1 - \beta_1} K_t g_t \\
&= -\frac{\alpha_t (1 - K_t)}{1 - \beta_1^t} g_t - \frac{\alpha_t K_t}{1 - \beta_1} g_t \\
&= -\frac{\alpha_t}{1 - \beta_1} g_t
\end{aligned}
\tag{70}
$$

Thus,

$$
\begin{aligned}
\|\xi_{t+1} - \xi_t\|_2^2 &= \left\| -\frac{\alpha_t (1 - K_t)}{1 - \beta_1} g_t - \frac{\alpha_t K_t}{1 - \beta_1} g_t \right\|_2^2 \\
&= \left(-\frac{\alpha_t}{1 - \beta_1}\right)^2 \|g_t\|_2^2 \\
&= \frac{\alpha_t^2}{(1 - \beta_1)^2} \|g_t\|_2^2 \\
&= \frac{\alpha_t^2}{(1 - \beta_1)^2} \sum_{i=1}^d g_{t,i}^2 \\
&\leq \frac{\alpha_t^2}{(1 - \beta_1)^2} \sum_{i=1}^d G_i^2
\end{aligned}
\tag{71}
$$

When $t \geq 2$,

$$
\begin{aligned}
\xi_{t+1} - \xi_t &= \theta_{t+1} + \frac{\beta_1}{1 - \beta_1} \left(\theta_{t+1} - \theta_t\right) - \theta_t - \frac{\beta_1}{1 - \beta_1} \left(\theta_t - \theta_{t-1}\right) \\
&= \frac{1}{1 - \beta_1} \left(\theta_{t+1} - \theta_t\right) - \frac{\beta_1}{1 - \beta_1} \left(\theta_t - \theta_{t-1}\right)
\end{aligned}
\tag{72}
$$

Due to

$$
\begin{aligned}
\theta_{t+1} - \theta_t &= -\alpha_t \widehat{g}_t \\
&= -\frac{\alpha_t (1 - K_t)}{1 - \beta_1^t} m_t - \alpha_t K_t g_t \\
&= -\frac{\alpha_t (1 - K_t)}{1 - \beta_1^t} \left(\beta_1 m_{t-1} + (1 - \beta_1) g_t\right) - \alpha_t K_t g_t
\end{aligned}
\tag{73}
$$

So,

$$
\begin{aligned}
&\xi_{t+1} - \xi_t \\
&= \frac{1}{1 - \beta_1} \left(-\frac{\alpha_t (1 - K_t)}{1 - \beta_1^t} \left(\beta_1 m_{t-1} + (1 - \beta_1) g_t\right) - \alpha_t K_t g_t\right) - \frac{\beta_1}{1 - \beta_1} \left(-\frac{\alpha_{t-1} (1 - K_{t-1})}{1 - \beta_1^{t-1}} m_{t-1} - \alpha_{t-1} K_{t-1} g_{t-1}\right) \\
&= -\frac{\beta_1}{1 - \beta_1} m_{t-1} \odot \left(\frac{\alpha_t (1 - K_t)}{1 - \beta_1^t} - \frac{\alpha_{t-1} (1 - K_{t-1})}{1 - \beta_1^{t-1}}\right) - \frac{\alpha_t (1 - K_t)}{1 - \beta_1^t} g_t - \frac{\alpha_t K_t}{1 - \beta_1} g_t + \frac{\beta_1}{1 - \beta_1} \alpha_{t-1} K_{t-1} g_{t-1} \\
&= -\frac{\beta_1}{1 - \beta_1} m_{t-1} \odot \left(\frac{\alpha_t (1 - K_t)}{1 - \beta_1^t} - \frac{\alpha_{t-1} (1 - K_{t-1})}{1 - \beta_1^{t-1}}\right) - \left(\frac{\alpha_t (1 - K_t)}{1 - \beta_1^t} + \frac{\alpha_t K_t}{1 - \beta_1}\right) g_t + \frac{\beta_1 \alpha_{t-1} K_{t-1}}{1 - \beta_1} g_{t-1}
\end{aligned}
\tag{74}
$$

We have:

$$\|\xi_{t+1} - \xi_t\|_2^2 \le 2 \left\| -\frac{\beta_1}{1 - \beta_1} m_{t-1} \odot \left( \frac{\alpha_t(1 - K_t)}{1 - \beta_1^t} - \frac{\alpha_{t-1}(1 - K_{t-1})}{1 - \beta_1^{t-1}} \right) \right\|_2^2$$

$$+ 2 \left\| -\left( \frac{\alpha_t(1 - K_t)}{1 - \beta_1^t} + \frac{\alpha_t K_t}{1 - \beta_1} \right) g_t \right\|_2^2 + 2 \left\| \frac{\beta_1 \alpha_{t-1} K_{t-1}}{1 - \beta_1} g_{t-1} \right\|_2^2$$

$$\le 2 \frac{\beta_1^2}{(1 - \beta_1)^2} \|m_{t-1}\|_\infty^2 \left\| \frac{\alpha_t(1 - K_t)}{1 - \beta_1^t} - \frac{\alpha_{t-1}(1 - K_{t-1})}{1 - \beta_1^{t-1}} \right\|_\infty \cdot \left\| \frac{\alpha_t(1 - K_t)}{1 - \beta_1^t} - \frac{\alpha_{t-1}(1 - K_{t-1})}{1 - \beta_1^{t-1}} \right\|_1$$

$$+ 2 \left\| -\left( \frac{\alpha_t(1 - K_t)}{1 - \beta_1^t} + \frac{\alpha_t K_t}{1 - \beta_1} \right) g_t \right\|_2^2 + 2 \left\| \frac{\beta_1 \alpha_{t-1} K_{t-1}}{1 - \beta_1} g_{t-1} \right\|_2^2 \tag{75}$$

Because

- $|m_{t-1,i}| = (1 - \beta_1^t) |\widehat{m}t, i| \le |\widehat{m}t, i| \le G_i, \|m_{t-1}\| \infty^2 \le (\max i G_i)^2$

- $\|g_t\|_2^2 = \sum_{i=1}^d g_{t,i}^2 \le \sum_{i=1}^d G_i^2$

- $K_t \in 0, 1^d$, we have $\|K_t\|_\infty \le \sum_{i=1}^d \mathbf{1}_i, \|1 - K_t\| \infty \le \sum_{i=1}^d \mathbf{1}_i \le d$

$$\alpha_t / \left( 1 - \beta_1^t \right) \ge 0, \ \alpha_{t-1} / \left( 1 - \beta_1^{t-1} \right) / \ge 0$$

$$\alpha_t \le \alpha_{t-1}, \ \frac{1}{1 - \beta_1^t} \le \frac{1}{1 - \beta_1^{t-1}}$$

$$\implies \frac{\alpha_t}{1 - \beta_1^t} \le \frac{\alpha_{t-1}}{1 - \beta_1^{t-1}}$$

$$\implies \left| \frac{\alpha_t}{1 - \beta_1^t} - \frac{\alpha_{t-1}}{1 - \beta_1^{t-1}} \right| \tag{76}$$

$$= \alpha_{t-1} / \left( 1 - \beta_1^{t-1} \right) - \alpha_t / \left( 1 - \beta_1^t \right)$$

$$\le \alpha_{t-1} / \left( 1 - \beta_1^{t-1} \right) \le \alpha_1 / \left( 1 - \beta_1 \right)$$

$$\implies \left\| \frac{\alpha_t (1 - K_t)}{1 - \beta_1^t} - \frac{\alpha_{t-1} (1 - K_{t-1})}{1 - \beta_1^{t-1}} \right\|_\infty \le \frac{\alpha_1}{(1 - \beta_1)}$$

$$\left\| \frac{\alpha_t (1 - K_t)}{1 - \beta_1^t} - \frac{\alpha_{t-1} (1 - K_{t-1})}{1 - \beta_1^{t-1}} \right\|_1 \le \sum_{i=1}^d \left( \alpha_{t-1} / \left( 1 - \beta_1^{t-1} \right) - \alpha_t / \left( 1 - \beta_1^t \right) \right) \mathbf{1}_i \le d \left( \alpha_{t-1} / \left( 1 - \beta_1^{t-1} \right) - \alpha_t / \left( 1 - \beta_1^t \right) \right) \tag{77}$$

Therefore

$$\|\xi_{t+1} - \xi_t\|_2^2 \le 2 \frac{\beta_1^2}{(1 - \beta_1)^2} \left( \max_i G_i \right)^2 \frac{d\alpha_1}{(1 - \beta_1)} \cdot \left( \frac{\alpha_{t-1}}{\left( 1 - \beta_1^{t-1} \right)} - \frac{\alpha_t}{\left( 1 - \beta_1^t \right)} \right) + 4 \frac{\alpha_t^2}{(1 - \beta_1)^2} \sum_{i=1}^d G_i^2 \tag{78}$$

**For term (3)**

When $t = 1$, referring to the case of $t = 1$ in the previous subsection,

$$\langle \nabla f(\theta_t), \xi_{t+1} - \xi_t \rangle = \left\langle \nabla f(\theta_t), -\frac{\alpha_t}{1 - \beta_1} g_t \right\rangle$$

$$= \left\langle \nabla f(\theta_t), -\frac{\alpha_t}{1 - \beta_1} \nabla f(\theta_t) \right\rangle + \left\langle \nabla f(\theta_t), -\frac{\alpha_t}{1 - \beta_1} \zeta_t \right\rangle \tag{79}$$

The last equality is due to the definition of $g_t$: $g_t = \nabla f(\theta_t) + \zeta_t$. Let's consider them separately:

$$
\begin{aligned}
\left\langle \nabla f(\theta_t), -\frac{\alpha_t}{1-\beta_1}\nabla f(\theta_t)\right\rangle &= -\frac{\alpha_t}{1-\beta_1}[\nabla f(\theta_t)][\nabla f(\theta_t)] \\
&\leq -\frac{\alpha_t}{1-\beta_1}\|\nabla f(\theta_t)\|_2^2
\end{aligned}
\tag{80}
$$

$$
\begin{aligned}
\left\langle \nabla f(\theta_t), -\frac{\alpha_t}{1-\beta_1}\zeta_t\right\rangle &\leq \frac{\alpha_t}{1-\beta_1}\|\nabla f(\theta_t)\|_2\|\zeta_t\|_2 \\
&= \frac{\alpha_t}{1-\beta_1}\|\nabla f(\theta_t)\|_2\|g_t - \nabla f(\theta_t)\|_2 \\
&\leq \frac{\alpha_t}{1-\beta_1}\cdot 2\sum_{i=1}^d G_i^2
\end{aligned}
\tag{81}
$$

Thus

$$
\begin{aligned}
&\langle \nabla f(\theta_t), \xi_{t+1} - \xi_t\rangle \\
&\leq -\frac{\alpha_t}{(1-\beta_1)}\|\nabla f(\theta_t)\|_2^2 + \frac{2\alpha_t}{1-\beta_1}\cdot\sum_{i=1}^d G_i^2
\end{aligned}
\tag{82}
$$

When $t \geq 2$,

$$
\begin{aligned}
\langle \nabla f(\theta_t), \xi_{t+1} - \xi_t\rangle &= \left\langle \nabla f(\theta_t), -\frac{\beta_1}{1-\beta_1}m_{t-1}\odot\left(\frac{\alpha_t(1-K_t)}{1-\beta_1^t} - \frac{\alpha_{t-1}(1-K_{t-1})}{1-\beta_1^{t-1}}\right)\right\rangle \\
&+ \left\langle \nabla f(\theta_t), -\left(\frac{\alpha_t(1-K_t)}{1-\beta_1^t} + \frac{\alpha_t K_t}{1-\beta_1}\right)\nabla f(\theta_t)\right\rangle + \left\langle \nabla f(\theta_t), -\left(\frac{\alpha_t(1-K_t)}{1-\beta_1^t} + \frac{\alpha_t K_t}{1-\beta_1}\right)\zeta_t\right\rangle \\
&+ \left\langle \nabla f(\theta_{t-1}), \frac{\beta_1\alpha_{t-1}K_{t-1}}{1-\beta_1}\nabla f(\theta_{t-1})\right\rangle + \left\langle \nabla f(\theta_{t-1}), \frac{\beta_1\alpha_{t-1}K_{t-1}}{1-\beta_1}\zeta_{t-1}\right\rangle
\end{aligned}
\tag{83}
$$

Start by looking at the first item after the equal sign:

$$
\begin{aligned}
&\left\langle \nabla f(\theta_t), -\frac{\beta_1}{1-\beta_1}m_{t-1}\odot\left(\frac{\alpha_t(1-K_t)}{1-\beta_1^t} - \frac{\alpha_{t-1}(1-K_{t-1})}{1-\beta_1^{t-1}}\right)\right\rangle \\
&\leq \frac{\beta_1}{1-\beta_1}\|\nabla f(\theta_t)\|_\infty\|m_{t-1}\|_\infty\cdot\left\|\frac{\alpha_t(1-K_t)}{1-\beta_1^t} - \frac{\alpha_{t-1}(1-K_{t-1})}{1-\beta_1^{t-1}}\right\|_1 \\
&\leq \frac{\beta_1}{1-\beta_1}\left(\max_i G_i\right)\left(\max_i G_i\right)\cdot\sum_{i=1}^d\left(\frac{\alpha_{t-1}}{(1-\beta_1^{t-1})} - \frac{\alpha_t}{(1-\beta_1^t)}\right)\mathbf{1}_i \\
&\leq \frac{\beta_1}{1-\beta_1}\left(\max_i G_i\right)\left(\max_i G_i\right)\cdot d\left(\frac{\alpha_{t-1}}{(1-\beta_1^{t-1})} - \frac{\alpha_t}{(1-\beta_1^t)}\right)
\end{aligned}
\tag{84}
$$

The second and third terms after the equal sign:

$$
\begin{aligned}
&\left\langle \nabla f(\theta_t), -\left(\frac{\alpha_t(1-K_t)}{1-\beta_1^t} + \frac{\alpha_t K_t}{1-\beta_1}\right)\nabla f(\theta_t)\right\rangle + \left\langle \nabla f(\theta_t), -\left(\frac{\alpha_t(1-K_t)}{1-\beta_1^t} + \frac{\alpha_t K_t}{1-\beta_1}\right)\zeta_t\right\rangle \\
&\leq -\frac{\alpha_t}{1-\beta_1^t}\|\nabla f(\theta_t)\|_2^2 + \left\langle \nabla f(\theta_t), -\frac{\alpha_t}{1-\beta_1^t}\zeta_t\right\rangle
\end{aligned}
\tag{85}
$$

The fourth and fifth terms after the equal sign:

$$\left\langle \nabla f\left(\theta_{t-1}\right), \frac{\beta_1 \alpha_{t-1} K_{t-1}}{1-\beta_1} \nabla f\left(\theta_{t-1}\right)\right\rangle + \left\langle \nabla f\left(\theta_{t-1}\right), \frac{\beta_1 \alpha_{t-1} K_{t-1}}{1-\beta_1} \zeta_{t-1}\right\rangle$$

$$\leq \frac{\beta_1 \alpha_{t-1}}{1-\beta_1} \left\|\nabla f\left(\theta_t\right)\right\|_\infty \left\|\nabla f\left(\theta_t\right)\right\|_\infty \left\|\mathbf{1}_i\right\|_1 + \frac{\beta_1 \alpha_{t-1}}{1-\beta_1} \left\|\nabla f\left(\theta_t\right)\right\|_\infty \left\|\zeta_t\right\|_\infty \left\|\mathbf{1}_i\right\|_1$$

$$\leq \frac{\beta_1 \alpha_{t-1}}{1-\beta_1} \left(\max_i G_i\right)\left(\max_i G_i\right) \sum_{i=1}^d \mathbf{1}_i + \frac{\beta_1 \alpha_{t-1}}{1-\beta_1}\left(\max_i G_i\right)\left(2\max_i G_i\right) \sum_{i=1}^d \mathbf{1}_i \tag{86}$$

$$\leq \frac{\beta_1 \alpha_{t-1}}{1-\beta_1} \left(\max_i G_i\right)\left(\max_i G_i\right) d + \frac{\beta_1 \alpha_{t-1}}{1-\beta_1}\left(\max_i G_i\right)\left(2\max_i G_i\right) d$$

Final:

$$\langle \nabla f\left(\theta_t\right), \xi_{t+1} - \xi_t\rangle$$

$$\leq \frac{\beta_1}{1-\beta_1}\left(\max_i G_i\right)\left(\max_i G_i\right) \cdot d\left(\frac{\alpha_{t-1}}{\left(1-\beta_1^{t-1}\right)} - \frac{\alpha_t}{\left(1-\beta_1^t\right)}\right) - \frac{\alpha_t}{\left(1-\beta_1^t\right)}\left\|\nabla f\left(\theta_t\right)\right\|_2^2$$

$$+ \frac{\beta_1 \alpha_{t-1}}{1-\beta_1}\left(\max_i G_i\right)\left(\max_i G_i\right) d + \frac{\beta_1 \alpha_{t-1}}{1-\beta_1}\left(\max_i G_i\right)\left(2\max_i G_i\right) d + \left\langle \nabla f\left(\theta_t\right), -\frac{\alpha_t}{1-\beta_1^t}\zeta_t\right\rangle \tag{87}$$

**Summarizing the results**

Let's start summarizing: when $t = 1$,

$$f\left(\xi_{t+1}\right) - f\left(\xi_t\right) \leq \frac{L}{2} \cdot 0 + L \cdot \frac{\alpha_t^2}{\left(1-\beta_1\right)^2} \sum_{i=1}^d G_i^2 - \frac{\alpha_t}{\left(1-\beta_1\right)}\left\|\nabla f\left(\theta_t\right)\right\|_2^2 + \frac{2\alpha_t}{1-\beta_1} \cdot \sum_{i=1}^d G_i^2 \tag{88}$$

Taking the expectation over the random distribution of $\zeta_1, \zeta_2, \ldots, \zeta_t$ on both sides of the inequality:

$$\mathbb{E}_t\left[f\left(\xi_{t+1}\right) - f\left(\xi_t\right)\right] \leq L \cdot \frac{\alpha_t^2}{\left(1-\beta_1\right)^2} \sum_{i=1}^d G_i^2 - \frac{\alpha_t}{\left(1-\beta_1\right)}\mathbb{E}_t\left\|\nabla f\left(\theta_t\right)\right\|_2^2 + \frac{2\alpha_t}{1-\beta_1} \cdot \sum_{i=1}^d G_i^2 \tag{89}$$

When $t \geq 2$,

$$f\left(\xi_{t+1}\right) - f\left(\xi_t\right)$$

$$\leq \frac{L}{2}\frac{\beta_1^2}{\left(1-\beta_1\right)^2}\alpha_{t-1}^2 \sum_{i=1}^d G_i^2 + L \cdot 2\frac{\beta_1^2}{\left(1-\beta_1\right)^2}\left(\max_i G_i\right)^2 \frac{d\alpha_1}{\left(1-\beta_1\right)} \cdot \left(\frac{\alpha_{t-1}}{\left(1-\beta_1^{t-1}\right)} - \frac{\alpha_t}{\left(1-\beta_1^t\right)}\right)$$

$$+ L \cdot 4\frac{\alpha_t^2}{\left(1-\beta_1\right)^2} \sum_{i=1}^d G_i^2 + \frac{\beta_1}{1-\beta_1}\left(\max_i G_i\right)\left(\max_i G_i\right) \cdot d\left(\frac{\alpha_{t-1}}{\left(1-\beta_1^{t-1}\right)} - \frac{\alpha_t}{\left(1-\beta_1^t\right)}\right)$$

$$- \frac{\alpha_t}{\left(1-\beta_1^t\right)}\left\|\nabla f\left(\theta_t\right)\right\|_2^2 + \frac{\beta_1 \alpha_{t-1}}{1-\beta_1}\left(\max_i G_i\right)\left(\max_i G_i\right) d + \frac{\beta_1 \alpha_{t-1}}{1-\beta_1}\left(\max_i G_i\right)\left(2\max_i G_i\right) d$$

$$+ \left\langle \nabla f\left(\theta_t\right), -\frac{\alpha_t}{1-\beta_1^t}\zeta_t\right\rangle \tag{90}$$

Taking the expectation over the random distribution of $\zeta_1, \zeta_2, \ldots, \zeta_t$ on both sides of the inequality:

$$\mathbb{E}_t \left[ f\left(\xi_{t+1}\right) - f\left(\xi_t\right) \right]$$

$$\leq \frac{L}{2} \frac{\beta_1^2}{\left(1-\beta_1\right)^2} \alpha_{t-1}^2 \sum_{i=1}^d G_i^2 + L \cdot 2 \frac{\beta_1^2}{\left(1-\beta_1\right)^2} \left(\max_i G_i\right)^2 \frac{d\alpha_1}{\left(1-\beta_1\right)} \cdot \left( \frac{\alpha_{t-1}}{\left(1-\beta_1^{t-1}\right)} - \frac{\alpha_t}{\left(1-\beta_1^t\right)} \right)$$

$$+ L \cdot 4 \frac{\alpha_t^2}{\left(1-\beta_1\right)^2} \sum_{i=1}^d G_i^2 + \frac{\beta_1}{1-\beta_1} \left(\max_i G_i\right) \left(\max_i G_i\right) \cdot d \left( \frac{\alpha_{t-1}}{\left(1-\beta_1^{t-1}\right)} - \frac{\alpha_t}{\left(1-\beta_1^t\right)} \right)$$

$$- \frac{\alpha_t}{\left(1-\beta_1^t\right)} \mathbb{E}_t \left\| \nabla f\left(\theta_t\right) \right\|_2^2 + \frac{\beta_1 \alpha_{t-1}}{1-\beta_1} \left(\max_i G_i\right) \left(\max_i G_i\right) d + \frac{\beta_1 \alpha_{t-1}}{1-\beta_1} \left(\max_i G_i\right) \left(2 \max_i G_i\right) d$$

$$+ \mathbb{E}_t \left\langle \nabla f\left(\theta_t\right), -\frac{\alpha_t}{1-\beta_1^t} \zeta_t \right\rangle$$

$$(91)$$

Since the value of $\theta_t$ is independent of $g_t$, they are statistically independent of $\zeta_t$:

$$\mathbb{E}_t \left[ \left\langle \nabla f\left(\theta_t\right), -\frac{\alpha_t}{1-\beta_1^t} \zeta_t \right\rangle \right]$$

$$= \mathbb{E}_t \left[ \left\langle -\frac{\alpha_t}{1-\beta_1^t} \nabla f\left(\theta_t\right), \zeta_t \right\rangle \right] \qquad (92)$$

$$= \left\langle -\frac{\alpha_t}{1-\beta_1^t} \mathbb{E}_t \left[ \nabla f\left(\theta_t\right) \right], \mathbb{E}_t[\zeta_t]^0 \right\rangle = 0$$

Summing up both sides of the inequality for $t = 1, 2, \ldots, T$:

• Left side of the inequality (can be reduced to maintain the inequality)

$$\sum_{t=1}^T \text{LHS of the inequality} = \sum_{t=1}^T \mathbb{E}_t \left[ f\left(\xi_{t+1}\right) - f\left(\xi_t\right) \right]$$

$$= \sum_{t=1}^T \mathbb{E}_t \left[ f\left(\xi_{t+1}\right) \right] - \mathbb{E}_t \left[ f\left(\xi_t\right) \right] \qquad (93)$$

$$= \sum_{t=1}^T \mathbb{E}_t \left[ f\left(\xi_{t+1}\right) \right] - \mathbb{E}_{t-1} \left[ f\left(\xi_t\right) \right]$$

$$= \mathbb{E}_T \left[ f\left(\xi_{T+1}\right) \right] - \mathbb{E}_0 \left[ f\left(\xi_1\right) \right]$$

Since $f\left(\xi_{T+1}\right) \geq \min_\theta f\left(\theta\right) = f\left(\theta^*\right)$, $\xi_1 = \theta_1$, and both are deterministic:

$$\sum_{t=1}^T \mathbb{E}_t \left[ f\left(\xi_{t+1}\right) - f\left(\xi_t\right) \right] \geq \mathbb{E}_T \left[ f\left(\theta^*\right) \right] - \mathbb{E}_0 \left[ f\left(\theta_1\right) \right] \qquad (94)$$

$$= f\left(\theta^*\right) - f\left(\theta_1\right)$$

• The right side of the inequality (can be enlarged to keep the inequality valid)

We perform a series of substitutions to simplify the symbols:

When $t > 2$,

1. $\frac{L}{2} \frac{\beta_1^2}{(1-\beta_1)^2} \alpha_{t-1}^2 \sum_{i=1}^d G_i^2 \triangleq C_1 \alpha_{t-1}^2$

2. $L \cdot 2 \frac{\beta_1^2}{(1-\beta_1)^2} \left(\max_i G_i\right)^2 \frac{d\alpha_1}{(1-\beta_1)} \cdot \left( \frac{\alpha_{t-1}}{\left(1-\beta_1^{t-1}\right)} - \frac{\alpha_t}{\left(1-\beta_1^t\right)} \right) \triangleq C_2 \left( \frac{\alpha_{t-1}}{\left(1-\beta_1^{t-1}\right)} - \frac{\alpha_t}{\left(1-\beta_1^t\right)} \right)$

3. $L \cdot 4 \frac{\alpha_t^2}{(1-\beta_1)^2} \sum_{i=1}^d G_i^2 \leq L \cdot 4 \frac{\alpha_t^2}{(1-\beta_1)^2} \sum_{i=1}^d G_i^2 \triangleq C_3 \alpha_t^2$

4. $\frac{\beta_1}{1-\beta_1}\left(\max_i G_i\right)\left(\max_i G_i\right)\cdot d\left(\frac{\alpha_{t-1}}{\left(1-\beta_1^{t-1}\right)}-\frac{\alpha_t}{\left(1-\beta_1^t\right)}\right)\triangleq C_4\left(\frac{\alpha_{t-1}}{\left(1-\beta_1^{t-1}\right)}-\frac{\alpha_t}{\left(1-\beta_1^t\right)}\right)$

5. $-\frac{\alpha_t}{\left(1-\beta_1^t\right)}\mathbb{E}_t\left[\left\|\nabla f\left(\theta_t\right)\right\|_2^2\right]\le-\alpha_t\mathbb{E}_t\left[\left\|\nabla f\left(\theta_t\right)\right\|_2^2\right]$

6. $\frac{\beta_1\alpha_{t-1}}{1-\beta_1}\left(\max_i G_i\right)\left(\max_i G_i\right)d+\frac{\beta_1\alpha_{t-1}}{1-\beta_1}\left(\max_i G_i\right)\left(2\max_i G_i\right)d\triangleq C_5\alpha_{t-1}$

When $t=1$,

1. $L\cdot\frac{\alpha_t^2}{\left(1-\beta_1\right)^2}\sum_{i=1}^d G_i^2\le L\cdot 4\frac{\alpha_t^2}{\left(1-\beta_1\right)^2}\sum_{i=1}^d G_i^2=C_3\alpha_t^2$

2. $-\frac{\alpha_t}{\left(1-\beta_1\right)}\mathbb{E}_t\left[\left\|\nabla f\left(\theta_t\right)\right\|_2^2\right]\le-\alpha_t\mathbb{E}_t\left[\left\|\nabla f\left(\theta_t\right)\right\|_2^2\right]$

3. $\frac{2\alpha_t}{1-\beta_1}\cdot\sum_{i=1}^d G_i^2\triangleq C_6\alpha_t$

After substitution,

$$
\begin{aligned}
\sum_{t=1}^T\text{RHS of the inequality} &\le \sum_{t=2}^T C_1\alpha_{t-1}^2+\sum_{t=1}^T C_3\alpha_t^2-\sum_{t=1}^T\alpha_t\mathbb{E}_t\left[\left\|\nabla f\left(\theta_t\right)\right\|_2^2\right]\\
&+\sum_{t=2}^T\left(C_2+C_4\right)\left(\frac{\alpha_{t-1}}{\left(1-\beta_1^{t-1}\right)}-\frac{\alpha_t}{\left(1-\beta_1^t\right)}\right)+\sum_{t=1}^T C_5\alpha_{t-1}+\sum_{t=1}^T C_6\alpha_t\\
&=\sum_{t=2}^T C_1\alpha_{t-1}^2+\sum_{t=1}^T C_3\alpha_t^2-\sum_{t=1}^T\alpha_t\mathbb{E}_t\left[\left\|\nabla f\left(\theta_t\right)\right\|_2^2\right]+\sum_{t=1}^T C_5\alpha_{t-1}+\sum_{t=1}^T C_6\alpha_t\\
&+\sum_{i=1}^d\left(C_2+C_4\right)\sum_{t=2}^T\left(\frac{\alpha_{t-1}}{\left(1-\beta_1^{t-1}\right)}-\frac{\alpha_t}{\left(1-\beta_1^t\right)}\right)\\
&=\sum_{t=2}^T C_1\alpha_{t-1}^2+\sum_{t=1}^T C_3\alpha_t^2-\sum_{t=1}^T\alpha_t\mathbb{E}_t\left[\left\|\nabla f\left(\theta_t\right)\right\|_2^2\right]+\sum_{t=1}^T C_5\alpha_{t-1}+\sum_{t=1}^T C_6\alpha_t\\
&+\sum_{i=1}^d\left(C_2+C_4\right)\left(\frac{\alpha_1}{\left(1-\beta_1\right)}-\frac{\alpha_T}{\left(1-\beta_1^T\right)}\right)\\
&\le\left(C_1+C_3+C_5+C_6\right)\sum_{t=1}^T\alpha_t^2-\sum_{t=1}^T\alpha_t\mathbb{E}_t\left[\left\|\nabla f\left(\theta_t\right)\right\|_2^2\right]+\sum_{i=1}^d\left(C_2+C_4\right)\frac{\alpha_1}{\left(1-\beta_1\right)}\\
&\le\left(C_1+C_3+C_5+C_6\right)\sum_{t=1}^T\alpha_t^2-\sum_{t=1}^T\alpha_t\mathbb{E}_t\left[\left\|\nabla f\left(\theta_t\right)\right\|_2^2\right]+\left(C_2+C_4\right)\frac{\alpha_1}{\left(1-\beta_1\right)}
\end{aligned}
\tag{95}
$$

Combining the results of scaling on both sides of the inequality:

$$
f\left(\theta^*\right)-f\left(\theta_1\right)\le\left(C_1+C_3+C_5+C_6\right)\sum_{t=1}^T\alpha_t^2-\sum_{t=1}^T\alpha_t\mathbb{E}_t\left[\left\|\nabla f\left(\theta_t\right)\right\|_2^2\right]+\left(C_2+C_4\right)\frac{\alpha_1}{\left(1-\beta_1\right)}
$$

$$
\implies\sum_{t=1}^T\alpha_t\mathbb{E}_t\left[\left\|\nabla f\left(\theta_t\right)\right\|_2^2\right]\le\left(C_1+C_3+C_5+C_6\right)\sum_{t=1}^T\alpha_t^2+f\left(\theta_1\right)-f\left(\theta^*\right)+\left(C_2+C_4\right)\frac{\alpha_1}{\left(1-\beta_1\right)}
\tag{96}
$$

Due to $\mathbb{E}_t\left[\|\nabla f\left(\theta_t\right)\|_2^2\right] = \mathbb{E}_{t-1}\left[\|\nabla f\left(\theta_t\right)\|_2^2\right]$,

$$\sum_{t=1}^T \alpha_t \mathbb{E}_t\left[\|\nabla f\left(\theta_t\right)\|_2^2\right] = \sum_{t=1}^T \alpha_t \mathbb{E}_{t-1}\left[\|\nabla f\left(\theta_t\right)\|_2^2\right]$$

$$\geq \sum_{t=1}^T \alpha_t \min_{t=1,2,\ldots,T} \mathbb{E}_{t-1}\left[\|\nabla f\left(\theta_t\right)\|_2^2\right] \qquad (97)$$

$$= \min_{t=1,2,\ldots,T} \mathbb{E}_{t-1}\left[\|\nabla f\left(\theta_t\right)\|_2^2\right] \sum_{t=1}^T \alpha_t$$

$$= \cdot \mathbb{E}\left(T\right) \cdot \sum_{t=1}^T \alpha_t$$

Then let $C_1 + C_3 + C_5 + C_6 \triangleq C_7$, $\underbrace{f\left(\theta_1\right) - f\left(\theta^*\right)}_{\geq 0} + \left(C_2 + C_4\right)\frac{\alpha_1}{(1-\beta_1)} \triangleq C_8$, therefore

$$\mathbb{E}\left(T\right) \cdot \sum_{t=1}^T \alpha_t \leq C_7 \sum_{t=1}^T \alpha_t^2 + C_8$$

$$\Longrightarrow \mathbb{E}\left(T\right) \leq \frac{C_7 \sum_{t=1}^T \alpha_t^2 + C_8}{\sum_{t=1}^T \alpha_t} \qquad (98)$$

Since $\alpha_t = \alpha/\sqrt{t}, \sum_{t=1}^T \frac{1}{t} \leq 1 + \log T$, we have:

$$\mathbb{E}(T) \leq \frac{C_7 \alpha^2 (\log T + 1) + C_8}{2\alpha\sqrt{T}} \qquad (99)$$

## D  DETAILED EXPERIMENTAL SUPPLEMENT

We performed extensive comparisons with other optimizers, including SGD Monro (1951), AdamKingma & Ba (2014), RAdamLiu et al. (2019) and AdamWLoshchilov & Hutter (2017). The experiments include: (a) image classification on CIFAR datasetKrizhevsky et al. (2009) with VGG Simonyan & Zisserman (2014), ResNet He et al. (2016) and DenseNet Huang et al. (2017), and image recognition with ResNet on ImageNet Deng et al. (2009).

### D.1  IMAGE CLASSIFICATION WITH CNNS ON CIFAR

For all experiments, the model is trained for 200 epochs with a batch size of 128, and the learning rate is multiplied by 0.1 at epoch 150. We performed extensive hyperparameter search as described in the main paper. Detailed experimental parameters we place in Tab. 5. Here we report both training and test accuracy in Fig. 7 and Fig. 8. SGDF not only achieves the highest test accuracy, but also a smaller gap between training and test accuracy compared with other optimizers.

Table 5: Hyperparameters used for CIFAR-10 and CIFAR-100 datasets.

| Optimizer | Learning Rate | $\beta_1$ | $\beta_2$ | Epochs | Schedule | Weight Decay | Batch Size | $\varepsilon$ |
|---|---|---|---|---|---|---|---|---|
| SGDF | 0.3 | 0.9 | 0.999 | 200 | StepLR | 0.0005 | 128 | 1e-8 |
| SGD | 0.1 | 0.9 | - | 200 | StepLR | 0.0005 | 128 | - |
| Adam | 0.001 | 0.9 | 0.999 | 200 | StepLR | 0.0005 | 128 | 1e-8 |
| RAdam | 0.001 | 0.9 | 0.999 | 200 | StepLR | 0.0005 | 128 | 1e-8 |
| AdamW | 0.001 | 0.9 | 0.999 | 200 | StepLR | 0.01 | 128 | 1e-8 |
| MSVAG | 0.1 | 0.9 | 0.999 | 200 | StepLR | 0.0005 | 128 | 1e-8 |
| AdaBound | 0.001 | 0.9 | 0.999 | 200 | StepLR | 0.0005 | 128 | - |
| Sophia | 0.0001 | 0.965 | 0.99 | 200 | StepLR | 0.1 | 128 | - |
| Lion | 0.00002 | 0.9 | 0.99 | 200 | StepLR | 0.1 | 128 | - |

Note: StepLR indicates a learning rate decay by a factor of 0.1 at the 150th epoch.

### D.2  IMAGE CLASSIFICATION ON IMAGENET

We experimented with a ResNet18 on ImageNet classification task. For SGD, we set an initial learning rate of 0.1, and multiplied by 0.1 every 30 epochs; for SGDF, we use an initial learning rate of 0.5, set $\beta_1 = 0.5$. Weight decay is set as $10^{-4}$ for both cases. To match the settings in Liu et al. (2019). Detailed experimental parameters we place in Tab. 6. As shown in Fig. 9, SGDF achieves an accuracy very close to SGD.

Table 6: Hyperparameters used for ImageNet.

| Optimizer | Learning Rate | $\beta_1$ | $\beta_2$ | Epochs | Schedule | Weight Decay | Batch Size | $\varepsilon$ |
|---|---|---|---|---|---|---|---|---|
| SGDF | 0.5 | 0.5 | 0.999 | 100 | StepLR | 0.0005 | 256 | 1e-8 |
| SGD | 0.1 | - | - | 100 | StepLR | 0.0005 | 256 | - |
| SGDF | 0.5 | 0.5 | 0.999 | 90 | Cosine | 0.0005 | 256 | 1e-8 |
| SGD | 0.1 | - | - | 90 | Cosine | 0.0005 | 256 | - |

Note: StepLR indicates a learning rate decay by a factor of 0.1 every 30 epochs.

### D.3  OBJECTIVE DETECTION ON PASCAL VOC

We show the results on PASCAL VOCEveringham et al. (2010). Object detection with a Faster-RCNN modelRen et al. (2015). Detailed experimental parameters we place in Fig. 7. The results are reported in Tab. 3, and detection examples shown in Fig. 10. These results also illustrate that our method is still efficient in object detection tasks.

Table 7: Hyperparameters for object detection on PASCAL VOC using Faster-RCNN+FPN with different optimizers.

| Optimizer | Learning Rate | $\beta_1$ | $\beta_2$ | Epochs | Schedule | Weight Decay | Batch Size | $\varepsilon$ |
|---|---|---|---|---|---|---|---|---|
| SGDF | 0.01 | 0.9 | 0.999 | 4 | StepLR | 0.0001 | 2 | 1e-8 |
| SGD | 0.01 | 0.9 | - | 4 | StepLR | 0.0001 | 2 | - |
| Adam | 0.0001 | 0.9 | 0.999 | 4 | StepLR | 0.0001 | 2 | 1e-8 |
| AdamW | 0.0001 | 0.9 | 0.999 | 4 | StepLR | 0.0001 | 2 | 1e-8 |
| RAdam | 0.0001 | 0.9 | 0.999 | 4 | StepLR | 0.0001 | 2 | 1e-8 |

Note: StepLR schedule indicates a learning rate decay by a factor of 0.1 at the last epoch.

### D.4 IMAGE GENERATION.

We experiment with one of the most widely used models, the Wasserstein-GAN with gradient penalty (WGAN-GP)Salimans et al. (2016) using a small model with a vanilla CNN generator. Using popular optimizerLuo et al. (2019); Zaheer et al. (2018); Balles & Hennig (2018); Bernstein et al. (2020), we train the model for 100 epochs, generate 64,000 fake images from noise, and compute the Frechet Inception Distance (FID)Heusel et al. (2017) between the fake images and real dataset (60,000 real images). FID score captures both the quality and diversity of generated images and is widely used to assess generative models (lower FID is better). For SGD and MSVAG, we report results from Zhuang et al. (2020). We perform 5 runs of experiments, and report the results in Fig. 4. Detailed experimental parameters we place in Tab. 8.

Table 8: Hyperparameters for Image Generation Tasks.

| Optimizer | Learning Rate | $\beta_1$ | $\beta_2$ | Epochs | Batch Size | $\varepsilon$ |
|---|---|---|---|---|---|---|
| SGDF | 0.01 | 0.5 | 0.999 | 100 | 64 | 1e-8 |
| Adam | 0.0002 | 0.5 | 0.999 | 100 | 64 | 1e-8 |
| AdamW | 0.0002 | 0.5 | 0.999 | 100 | 64 | 1e-8 |
| Fromage | 0.01 | 0.5 | 0.999 | 100 | 64 | 1e-8 |
| RMSProp | 0.0002 | 0.5 | 0.999 | 100 | 64 | 1e-8 |
| AdaBound | 0.0002 | 0.5 | 0.999 | 100 | 64 | 1e-8 |
| Yogi | 0.01 | 0.9 | 0.999 | 100 | 64 | 1e-8 |
| RAdam | 0.0002 | 0.5 | 0.999 | 100 | 64 | 1e-8 |

### D.5 EXTENDED EXPERIMENT.

The study involves evaluating the vanilla Adam optimization algorithm and its enhancement with a Wiener filter on the CIFAR-100 dataset. Fig. 11 contains detailed test accuracy curves for both methods across different models. The results indicate that the adaptive learning rate algorithms exhibit improved performance when supplemented with the proposed first-moment filter estimation. This suggests that integrating a Wiener filter with the Adam optimizer may improve performance.

### D.6 OPTIMIZER TEST.

We derived a correction factor $(1 - \beta_1)(1 - \beta_1^{2t})/(1 + \beta_1)$ from the geometric progression to correct the variance of by the correction factor. So we test the SGDF with or without correction in VGG, ResNet, DenseNet on CIFAR. We report both test accuracy in Fig. 12. It can be seen that the SGDF with correction exceeds the uncorrected one.

We built a simple neural network to test the convergence speed of SGDF compared to SGDM and vanilla SGD. We trained 5 epochs and recorded the loss every 30 iterations. As Fig. 13 shown, the convergence rate of the filter method surpasses that of the momentum method, which in turn exceeds that of vanilla SGD.

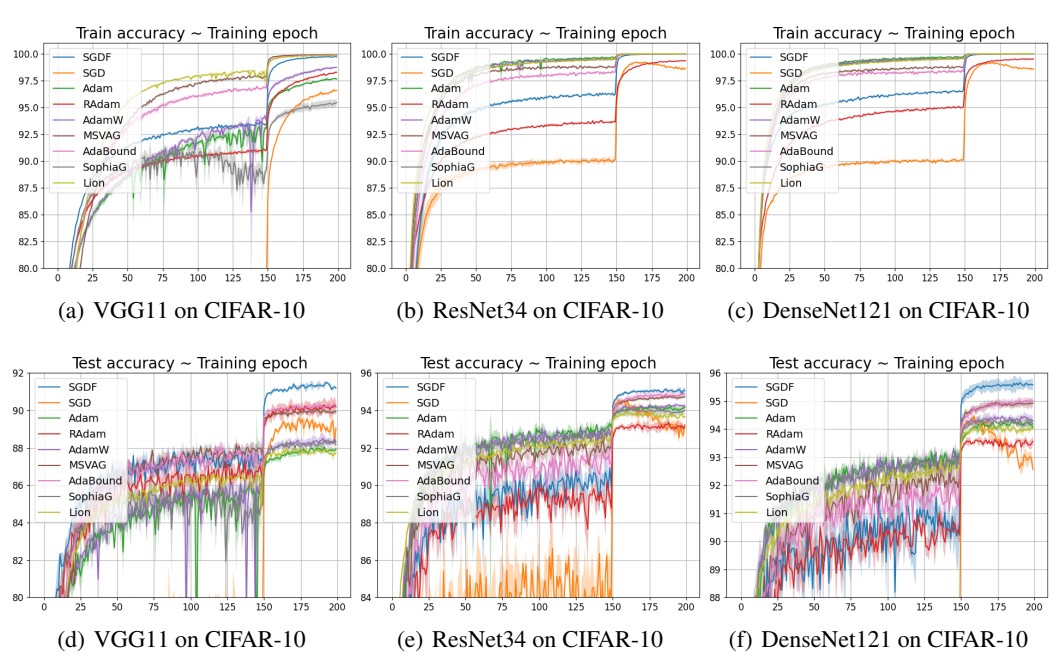

Figure 7: Training (top row) and test (bottom row) accuracy of CNNs on CIFAR-10 dataset. We report confidence interval ($[\mu \pm \sigma]$) of 3 independent runs.

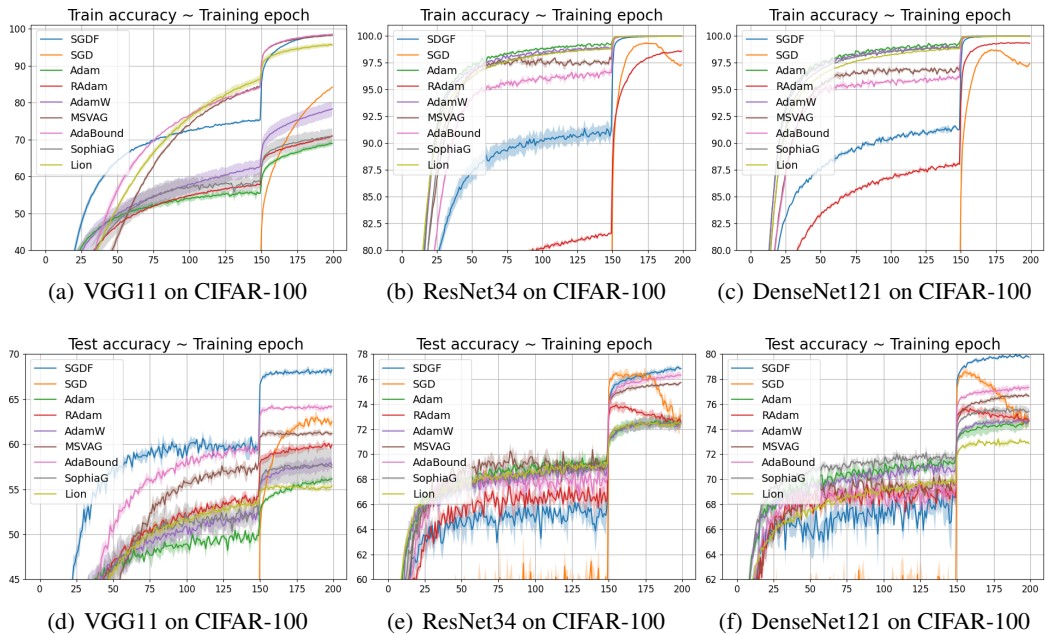

Figure 8: Training (top row) and test (bottom row) accuracy of CNNs on CIFAR-100 dataset. We report confidence interval ($[\mu \pm \sigma]$) of 3 independent runs.

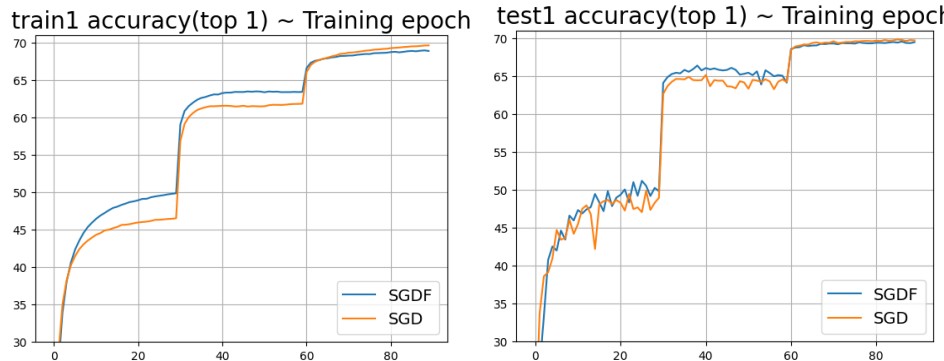

Figure 9: Training and test accuracy (top-1) of ResNet18 on ImageNet.

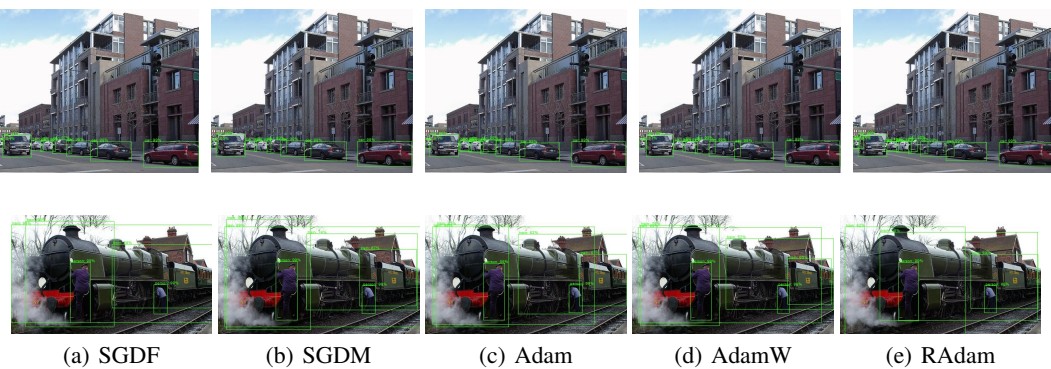

(a) SGDF      (b) SGDM      (c) Adam      (d) AdamW      (e) RAdam

Figure 10: Detection examples using Faster-RCNN + FPN trained on PASCAL VOC.

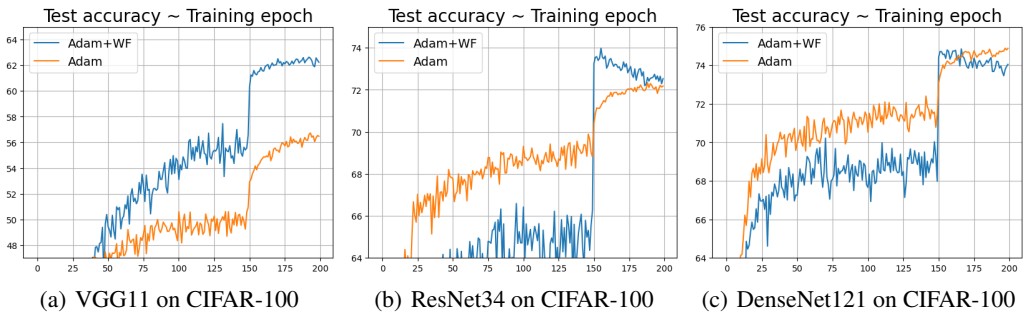

(a) VGG11 on CIFAR-100      (b) ResNet34 on CIFAR-100      (c) DenseNet121 on CIFAR-100

Figure 11: Test accuracy of CNNs on CIFAR-100 dataset. We train vanilla Adam and Adam combined with Wiener Filter.

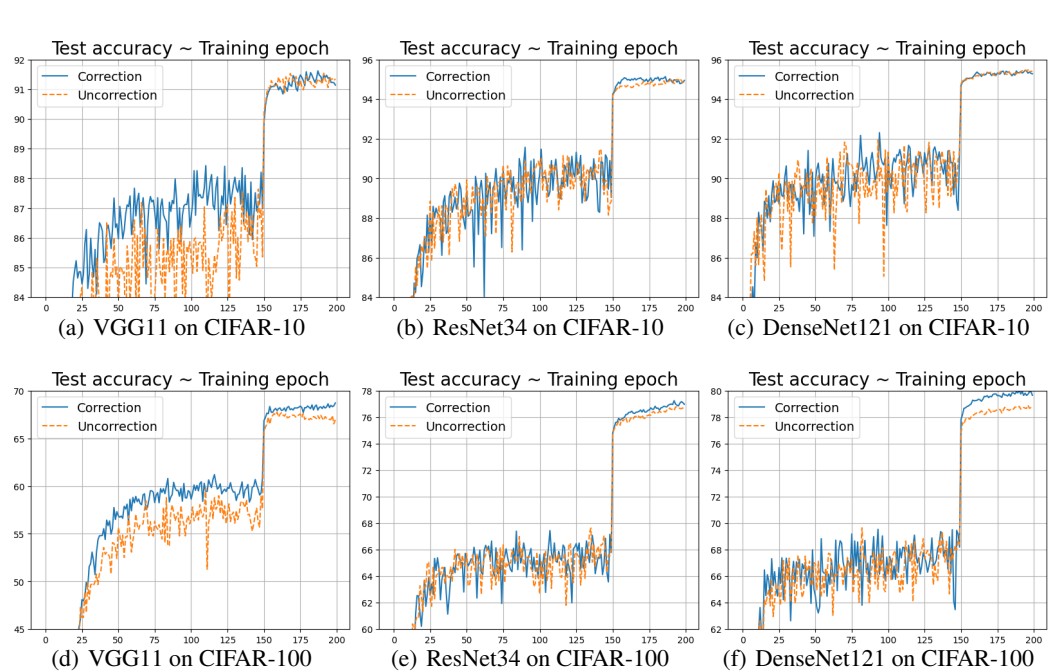

Figure 12: SGDF with or without the correction factor. The curve shows the accuracy of the test.

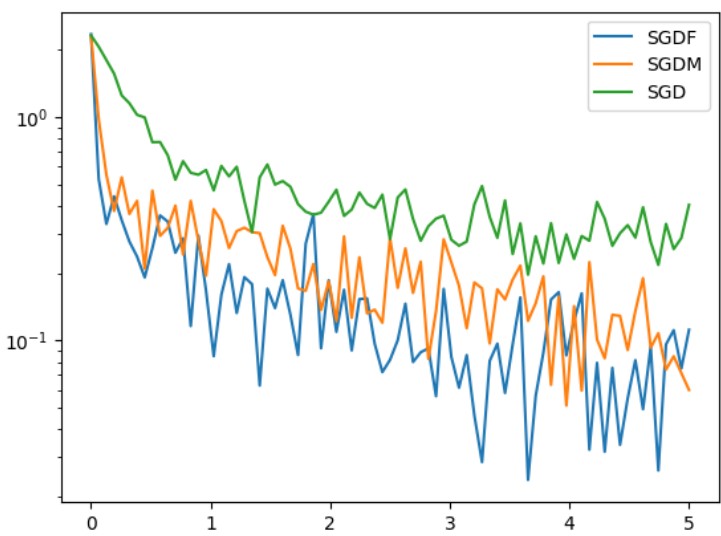

Figure 13: Comparison of convergence rates.

