# OpenReview forum: "SGDF: A Method for Reducing Variance in Stochastic Gradient Descent via Filter Estimation"
_ICLR.cc/2025/Conference — ICLR 2025 Conference Withdrawn Submission_

### Official Review · Reviewer_gbVo · 2024-11-02

**Soundness:** 3
**Presentation:** 2
**Contribution:** 3
**Rating:** 5
**Confidence:** 4

**Summary:**

This work presents a new algorithm for training deep neural networks by incorporating the Wiener filter scheme into SGD with momentum estimation. Theoretically, it presents some insight into the generalization and convergence analysis, while in the experiments, it also shows some numerical benefits in these regards.

**Strengths:**

The combination between deep learning optimization and signal processing theory is interesting.

**Weaknesses:**

1. How the Wiener filter is combined and derived is unclear in the main paper, though some derivations are given in the supplements. This is very important for the community in DL to understand the connections to signal processing theory and how it can be benefited, as it is the core idea behind the picture. In general, many elements about the proposed algorithm need further clarification.

2. It claimed the benefits of fast convergence in Sec 2 and Sec 3, which plays a key role in displaying the contribution, however, it is not reflected with numerical evidence nor explained well, despite a simple plot in the end of the supplements.

**Questions:**

1. Mainly related to Weakness1:

(a) In line 60 and 195, the claimed benefit of the gradient’s variance estimate are not well explained on how and why in SDGF.

(b) In line 106 and 107, I didn’t find a distinctive evidence on the faster and stable convergence in Fig. 1, or the connection to gradient smoothness.

(c) In line 121 and 122,  I didn’t find a distinctive (numerical?) evidence about less noise and no gradient shift on fig. 2(b), especially compared to fig. 2(c). And, why tilde peaks are shown on  fig. 2(b)?

(d), In sec 3.1, rather than the plain inline context explanations, I would suggest to specifically organize some derivations or some schematic explanations on connecting Wiener filter with Algorithm 1.

(e) Eqn. (1) and. (2) seem like a general results when computing the joint pdf of two independent distributions, which is therefore not peculiar to the presented method. How the claims in line 220 and 221 are not well elaborated.

(f) The unimpaired generalization ability as claimed by the authors for the proposed algorithm is not well reflected in sec 3.3, if I didn’t miss some details.

(g) In line 292 and 293, what if the terms related to the estimated gain $K_t$ are not scaled to their maximum upper bounds, what results can be expected?


2. Mainly related to experiments:

(a) I’m not sure if the comparison results are fairly solid, as Table 5, 6, 8 only show a single set of hyperparameters for different algorithms which for sure need different settings for their own best best performance. Could the authors explain the reasons and results more comprehensively?

 (b) In Fig.3, do they all use BN, how, and why and why not?

 (c) Adadelta and adagrad are also quite related.

 (d) why only SGD is compared in Table 2?

(e) In general, the explanations and results about fig. 5 and fig. 6 are insufficiently informative for presenting the proposed SDGF. For instance, why Adam has such results in fig. 5(d)? Can the sharpeness in Fig. 6(a) be different if the proposed SGDF is combined with Adam as in sec 4.4.?

(f) In related works, the SWA [1] and TWA [2] algorithms (and other related ones in this series) should also be mentioned, especially regarding the sharp and flat solutions.

3. See weakness 2

BTW, the titles in the paper and system should be unified.

[1] P. Izmailov, D. Podoprikhin, T. Garipov, D. Vetrov, and A. G Wilson, Averaging weights leads to wider optima and better generalization, UAI 2018.
[2] T. Li, Z. Huang, Q. Tao, Y. Wu, and X. Huang, Trainable weight averaging: Efficient training by optimizing historical solutions,  ICLR, 2023.

---

### Official Review · Reviewer_uGT1 · 2024-11-04

**Soundness:** 3
**Presentation:** 3
**Contribution:** 3
**Rating:** 5
**Confidence:** 2

**Summary:**

This paper introduces a method, SGDF, which enhances SGDM with a new Exponential Moving Variance sequence. By incorporating an exponential moving average between the gradient and its momentum, this method addresses noisy optimization scenarios, resulting in more stable convergence compared to SGDM. The authors provide theoretical justification for their results and validate them through extensive experiments.

**Strengths:**

**S1:** The authors analyze the convergence properties of SGDF in convex and non-convex cases.

**S2:** The authors conduct experimental validation on a wide range of problems and provide detailed descriptions of their baselines.

**S3:** The authors provide a visualization of loss landscapes for the main baseline method.

**S4:** Methodology behind the main algorithm extends to Adam modifications, which outperform the Vanilla-Adam baselines.

**Weaknesses:**

Overall, this paper has the potential to make a significant algorithmic contribution. However, I have noted several concerns and kindly request clarifications from the authors:

**W1:** In Lines 46-47, the authors state that SGD with Filter was proposed in [1]. Could you provide more detailed commentary on the relationship between this method and the algorithm from [1]?

**W2:** The authors introduce a new algorithm – Wiener Adam (Sec. 4.4). It would benefit the readers if the algorithm's pseudocode were included in the paper.

**W3:** In Line 364, the authors refer to using the best-reported parameters from [2, 3]. Could you clarify which specific parameters were utilized? I noticed that the model architectures used for training on the CIFAR-100 dataset differ: {VGG16, ResNet18, WideResNet} are used in Table 4 of [2], while {VGG11, ResNet32, DenseNet21} are used in this paper. The only alignment with [2] is the SGD (with Momentum) result for ResNet18 on the ImageNet task. Additionally, [3] does not include any computer vision tasks.

**W4:** The experimental findings of the paper are based exclusively on image datasets. I kindly request that the authors include at least one NLP experiment to demonstrate that SGDF can effective beyond convolutional-based functions.

**W5:** In Figure 2, the authors highlight the differences between (b) and (c) histograms and conclude (Line 120) that, “Compared to other methods, SGDF produces a gradient distribution as concentrated as BN.” Could you clarify this part of the manuscript in more details, since (b) and (c) are almost the same.

**W6:** Through the text, e.g., in Line 159 an important thins may occur and left uncovered. In this case, the list of assumptions before the main lemmas and theorems could improve the flow of the entire paper.

**W7:** For novelty, it is unclear if Theorem3.1 and A.1-A.3,  stated as novel results?

**W8:** In Lines 318-330, the authors describe a hyperparameters sweep. Why the value for SGDF's learning rate is missing? Please, provide a missing values.

**W9:**  In Lines 482-489, the authors discuss the effect of BatchNorm on the overall performance. They note, that for deep learning models with BatchNorm, the benefit of using Wiener Adam is not significant. Could the authors comment on how performance might change if LayerNorm is used instead of BatchNorm? This question is motivated by the use of LayerNorm in ViTs and Language Models.


**Minor Comments:**

**C1:** In Line 25, I would kindly suggest to write ''the optimizer serves as s critical component of the training process''.



[1] Stuart Geman, et al., ''Neural networks and the bias/variance dilemma. Neural Computation''.

[2] Jinghui Chen, et al., ''Closing the Generalization Gap of Adaptive Gradient Methods in Training Deep Neural Networks''.

[3] Liyuan Liu, et al., ''On the Variance of the Adaptive Learning Rate and Beyond''. ICLR 2020

**Questions:**

See Weaknesses.

---

### Official Review · Reviewer_1p7N · 2024-11-04

**Soundness:** 2
**Presentation:** 2
**Contribution:** 2
**Rating:** 3
**Confidence:** 5

**Summary:**

The paper introduces SGDF (SGD with Filter), a new optimization approach designed to accelerate the convergence of SGD while maintaining generalization quality. The method leverages Wiener filter theory to reduce the noise associated with SGD, thereby improving the first-order moment estimation. By introducing a time-varying adaptive gain, SGDF seeks to balance the efficiency of convergence with robust generalization. Empirical results suggest that SGDF competes favorably with other state-of-the-art optimizers.

**Strengths:**

1. The application of Wiener filter theory to reduce noise in SGD is novel and shows a unique analytical approach to improving optimization.
2. Empirical results demonstrate competitive performance compared to existing optimization methods.

**Weaknesses:**

The arguments presented lack rigor. Here are some examples.
   1) In Figure 2, where the gradient norms of various optimizers are plotted. This section is unclear, as the update norms might be more indicative of performance rather than gradient norms, raising questions about the relevance of the data presented.
  2) In Section 3.3, the paper lacks a convincing explanation for how Theorem 3 supports the generalization claim. How is the $f(\theta)$, $D_{i,j}$, $P(\theta)$ related with different optimizers?
 3) The convergence analysis relies on highly restrictive assumptions, such as bounded gradients and convexity, whereas recent research has relaxed these to more practical assumptions like affine noise or relaxed smoothness.
 4) Appendix A's proof is unclear, particularly in the definitions of variables \( T \) and \( t \), and the relationship between them. This ambiguity complicates the interpretation of the results presented.

The experimental results are not significant, please see Table 1 and Table 2. Moreover, the algorithms are run for old tasks (cifar + ResNet), which are not interesting for the community where the SOTA models are all of Transformer-like architectures.

**Questions:**

How does the algorithm perform on Transformer-like models?

---

### Official Review · Reviewer_G6ay · 2024-11-04

**Soundness:** 1
**Presentation:** 1
**Contribution:** 2
**Rating:** 3
**Confidence:** 3

**Summary:**

The paper uses Wiener filter theory to reduce the variance of the historical gradient by using a time varying adaptive gain. The paper also demonstrates the effectiveness of SGD with the new Wiener filter. The paper also points out to code.

**Strengths:**

1) Novelty : Proposes a new optimiser SGDF.
2) Significance : If the improved performance of the optimiser is demonstrated in a sound manner then it is a good addition.
3) Soundness in Experiments: The new algorithm is certain run on a decent enough range of standard datasets and model. However, the newer method outperforms prior methods in only Table 3, and the improvements over prior methods is very marginal in Table 1 and 2 and Figure 4.
4) Clarity : The paper is marginal in terms of clarity.

**Weaknesses:**

The paper suffers mainly soundness and clarity.

Soundness
1) Variance is never defined at all
2) Fusion is never defined at all
3) What the connection between of frequency response which arises in signal process and linear systems theory and gradient descent? What is the meaning of frequency response in the context of optimisation of deep neural networks?
4) There are several claims which seemed to be made in the passing in the text that explains Figure 1 and 2.
 (a) VGG without BN has "significant" instability. How was instability measure? what will be an insignificant amount of instability in this measure? and why with respect to this measure the VGG without BN has `significant' instability?
 (b) network oscillation and difficult to converge stably : by instability do we mean iterates running off to infinity? It is hard to reason out why oscillation is a bad thing, as a matter of fact, in the quadratic loss case, the optimal learning rate is 2/(maximum_eigenvalue + minimum_eigenvalue), and for this choice of learning rate, the parameter corresponding to top eigenvalue component oscillates. So, it is hard to understand why oscillation is a bad thing.
(c) in Figure 2(d) how do we know that the drift in the SGD-LPF case is due to the fixed weighting coefficient?
(d) SGDF produces gradients with less noise (how was noise measured? can it be indicated by a number?) and no gradient shift (how was gradient shift measured, can it be indicated by a number?)
5) What is fusion of Gaussian distributions? Line 200 mentions that the Wiener filter fuses the gradient and the momentum estimate? Is fusion is simple addition (in the case of adding two independent random variable the corresponding probability density functions convolve and not multiply).
6) What does the Fokker-Planck equation correspond to in the paper? What is n in equation (3) in Theorem 3.1.
7) The variance matrix D is not defined? Which quantities variances does D denote?
8) Poor choice of comparator: https://github.com/weiaicunzai/pytorch-cifar100, here VGGs have maximum of 31% error, i.e., 69% or more performance. However, the optimisers in Figure 1 including SGDF are worser than the numbers in https://github.com/weiaicunzai/pytorch-cifar100.
9) How did $$\hat{m}_t$$ and $$g_t$$ in equation (13) become $$x$$ in equation (14).

Clarity
1) Please define fusion, variance D and clarify how frequency response of the Wiener filter is relevant to optimisation.

**Questions:**

If signal preservation is the only main issue, then are we saying that GD (with gradient calculated on the entire dataset instead of mini-batches) will outperform all the other methods?

---

### Official Review · Reviewer_h2yv · 2024-11-08

**Soundness:** 3
**Presentation:** 3
**Contribution:** 1
**Rating:** 5
**Confidence:** 4

**Summary:**

This paper presents SGDF (Stochastic Gradient Descent with Filter), which enhances SGD by integrating Wiener filter theory from signal processing. SGDF utilizes a time-varying adaptive gain and applies a filter to historical gradients to improve first-moment estimation. Empirical results on CIFAR and ImageNet suggest that SGDF performs competitively with popular optimizers like Adam, demonstrating robust convergence properties and generalization.

**Strengths:**

- The motivation for combining SGD with filter theory is well-articulated, emphasizing the need for faster convergence without sacrificing generalization. The derivation of the new method is novel.
- This paper is clearly written and easy to follow.

**Weaknesses:**

- SGDF does not achieve SOTA performance on several benchmarking tasks, such as ResNet34 on CIFAR10, DenseNet121 on CIFAR10, AdaBelief reports better results in their work. "AdaBelief Optimizer: Adapting Stepsizes by the Belief in Observed Gradients". Its hard to convince practitioners to use this optimizer.
- The theoretic part of this paper provide standard results, i.e., similar convergence guarantees as the existing work. There is no clear theoretic contribution of this work.

**Questions:**

Could author elaborate the key benefit of SGDF over existing methods such as AdaBelief?

---

### Note · Authors · 2024-11-15

**Comment:**

We thank the reviewer for the detailed comments on our work. However, we did not allow this work to be clearly expressed due to presentation issues, so it was withdrawn.

**Withdrawal Confirmation:**

I have read and agree with the venue's withdrawal policy on behalf of myself and my co-authors.